# LncRNA *Ctcflos* orchestrates transcription and alternative splicing in thermogenic adipogenesis

Andrea Bast-Habersbrunner[1,2] , Christoph Kiefer[1] , Peter Weber[3], Tobias Fromme[1] , Anna Schießl[1], Petra C Schwalie[4] , Bart Deplancke[4] , Yongguo Li[1,2,*,†] & Martin Klingenspor[1,2,**,†]

## Abstract

**The recruitment of thermogenic brite adipocytes within white adipose tissue attenuates obesity and metabolic comorbidities, arousing interest in understanding the underlying regulatory mechanisms. The molecular network of brite adipogenesis, however, remains largely unresolved. In this light, long noncoding RNAs (lncRNAs) emerged as a versatile class of modulators that control many steps within the differentiation machinery. Leveraging the naturally varying propensities of different inbred mouse strains for white adipose tissue browning, we identify the nuclear lncRNA *Ctcflos* as a pivotal orchestrator of thermogenic gene expression during brite adipocyte differentiation. Mechanistically, *Ctcflos* acts as a pleiotropic regulator, being essential for the transcriptional recruitment of the early core thermogenic regulatory program and the modulation of alternative splicing to drive brite adipogenesis. This is showcased by *Ctcflos*-regulated gene transcription and splicing of the key browning factor *Prdm16* toward the isoform that is specific for the thermogenic gene program. Conclusively, our findings emphasize the mechanistic versatility of lncRNAs acting at several independent levels of gene expression for effective regulation of key differentiation factors to direct cell fate and function.**

**Keywords** brite adipocytes; long noncoding RNAs; Prdm16; splicing; Ucp1-dependent thermogenesis
**Subject Categories** Metabolism; RNA Biology

## Introduction

Continuously increasing adiposity and in particular excess visceral white fat accumulation in humans worldwide constitutes the major risk factor for several metabolic diseases including type 2 diabetes, hypertension, and cardiovascular diseases (Mann, 2002; Lloyd-Jones *et al*, 2009). For more than a century, several attempts have been made to develop pharmacological obesity treatments that, however, in many cases revealed to be unsafe, of moderate efficiency or short-lasting (reviewed in Müller *et al*, 2018). In 2009, with the discovery of considerable amounts of brown adipocytes in adult humans (Saito *et al*, 2009; van Marken Lichtenbelt *et al*, 2009; Virtanen *et al*, 2009), recruitment and activation of these cells evolved as a promising strategy that could complement existing therapies to achieve a more sustainable weight loss and to relieve obesity-associated comorbidities. As opposed to white adipocytes that store excess energy in the form of triglycerides, brown adipocytes dispense nutritional energy as heat via uncoupling protein 1 (UCP1). UCP1 is a transmembrane protein located in the inner membrane of brown fat mitochondria. Upon activation by lipolytically released free fatty acids, it allows proton flux from mitochondrial intermembrane space into matrix to mediate short-circuiting of the mitochondrial proton gradient that normally powers adenosine triphosphate (ATP) synthesis (Klingenspor, 2003). Compensatory substrate catabolism is enforced to fuel the respiratory chain in the attempt to re-establish the electrochemical gradient. Chemical energy of fatty acids and glucose is thus converted into thermal energy, making thermogenic fat a potent modulator in energy homeostasis in terms of energy expenditure, blood glucose, and lipid control in mouse and humans (Cederberg *et al*, 2001; Cypess *et al*, 2009; Bartelt *et al*, 2011; Bordicchia *et al*, 2012; Ouellet *et al*, 2012; Hanssen *et al*, 2015). Furthermore, prandial secretin-mediated brown fat activation induces meal-associated thermogenesis and consequently meal termination, attributing brown fat a role on both sides of the energy balance, in energy intake and energy expenditure (Li *et al*, 2018). In humans, the mass of BAT, however, is relatively small, limiting its contribution to whole body energy expenditure and metabolic homeostasis (reviewed in Carpentier *et al*, 2018).

In this regard, it is of great interest that thermogenic, UCP1 expressing fat cells can also be recruited within white adipose tissue (WAT) by acute cold exposure or β-adrenergic receptor activation in a process called browning (Young *et al*, 1984; Cousin *et al*, 1992;

1 Chair for Molecular Nutritional Medicine, TUM School of Life Sciences Weihenstephan, Technical University of Munich, Freising, Germany
2 EKFZ - Else Kröner-Fresenius Center for Nutritional Medicine, Technical University of Munich, Freising, Germany
3 Research Unit Radiation Cytogenetics, Helmholtz Center Munich Research Center for Environmental Health (GmbH), Neuherberg, Germany
4 School of Life Sciences, EPFL and Swiss Institute of Bioinformatics, Lausanne, Switzerland
*Corresponding author. Tel: +49 816 71 2368; E-mail: yongguo.li@tum.de
**Corresponding author (lead contact). Tel: +49 816 71 2386; E-mail: mk@tum.de
†These authors contributed equally to this work as senior authors

Guerra *et al*, 1998; Himms-Hagen *et al*, 2000). These brown-in-white (brite) or beige adipocytes share a similar UCP1-mediated thermogenic potential as classical brown adipocytes but clearly differ in origin, development, anatomical location, and gene signature (Atit *et al*, 2006; Morrison *et al*, 2008; Kajimura *et al*, 2010; Wu *et al*, 2012; Harms & Seale, 2013). The particularity of brite adipocytes to develop within WAT can be employed to redirect the main function of the organ from mere energy storage toward energy dissipation, thereby increasing the mass of thermogenic adipocytes, counteracting obesity, and improving metabolic health (Guerra *et al*, 1998; Cederberg *et al*, 2001; Seale *et al*, 2011; Bordicchia *et al*, 2012). Mapping of cold-inducible brown and brite adipocytes in lean and obese humans revealed a previously underestimated potential for the recruitment of thermogenic adipocytes in non-subcutaneous adipose tissue depots (Leitner *et al*, 2017).

Facing that goal, it is of pivotal interest to deepen our yet superficial knowledge on the regulatory mechanisms underlying WAT browning. Some transcriptional (co-) factors including peroxisome proliferator-activated receptor γ (PPARγ), CCAAT/enhancer-binding protein β (C/EBPβ), Pparγ coactivator 1α (PGC1α), and PR domain containing 16 (PRDM16) have been identified to be essential for brite adipogenesis (Harms & Seale, 2013; Chu & Gawronska-Kozak, 2017; Pradhan *et al*, 2017). PRDM16 emerged as a cell-autonomous key regulator controlling early determination and differentiation of thermogenic fat cells while repressing myogenic and white adipocyte lineages (Seale *et al*, 2011). Adipose tissue-specific ectopic expression and knockout of *Prdm16* demonstrate that it is sufficient and required for browning of WAT with impact on whole body metabolism (Seale *et al*, 2011; Cohen *et al*, 2014). These core transcription factors, however, sketch only a rough frame of the regulatory network in brite adipogenesis that needs to be further specified in more depth (Li *et al*, 2019a).

In this respect, recent advances in global sequencing technologies uncovered a new class of RNAs, long noncoding RNAs (lncRNAs), as powerful modulators of cellular differentiation. Rapid progress in whole-genome and whole-transcriptome profiling revealed that the human genome is pervasively transcribed but only 1.2% of it are protein-coding exons (Scheideler, 2019). This discrepancy weakened the former perception of RNA as mere messenger between DNA and protein and led to the discovery of further regulatory non-protein-coding RNA classes, including long noncoding RNAs (lncRNAs). Longtime disregarded as transcriptional junk, they nowadays emerge as versatile regulators and drivers of organismal complexity. LncRNAs are low abundant transcripts of more than 200 nucleotides length with highly tissue and developmental stage-specific expression fostering their regulatory potential in cellular differentiation (Derrien *et al*, 2012). Accordingly, some lncRNAs have been identified as critical components in white and brown adipogenesis (Sun *et al*, 2013; Zhao *et al*, 2014; Alvarez-Dominguez *et al*, 2015; Schmidt *et al*, 2018). Knowledge on the role of lncRNAs in brite adipogenesis, in contrast, remains limited and demands for a systematic approach to identify long noncoding transcripts of functional importance in differentiation and thermogenesis of these cells.

To this end, comparative studies between mouse strains with naturally varying browning propensities provide a fruitful approach that allows to fade out general noise unveiling the key factors that determine the capacity for WAT browning (Li *et al*, 2019a). Different inbred mouse strains possess diverging capacities to recruit

brite adipocytes in WAT in response to cold or β-adrenergic receptor stimulation (Koza *et al*, 2000; Xue *et al*, 2005). As mirrored by the expression level of *Ucp1*, SWR/J and BL/6J strains exhibit a low, while A/J, AKR/J, and 129S6 strains have a high browning propensity. This disparity is also maintained on the level of primary adipocytes *in vitro* in the absence of systemic influences including sympathetic innervation, endocrine stimuli, or the degree of angiogenesis and is independent of paracrine and autocrine signals. The propensity for brite adipocyte recruitment in WAT thus depends on cues intrinsic to the progenitor cells determined by their respective genetic background (Li *et al*, 2014).

Previously, we took advantage of this phenomenon within a systems-genetics approach to identify and validate several novel browning regulating protein-coding genes and to integrate them into a comprehensive regulatory network that orchestrates brite adipocyte differentiation (Li *et al*, 2019a). Here, we take advantage of these resources in a comparative transcriptome analysis to complement the established browning regulatory network of protein-coding genes with lncRNAs as a further layer of regulation. Among the identified lncRNAs, *Ctcflos* stood out as the top candidate that acts as a novel essential regulator of differentiation, maintenance, and thermogenic function of brite adipocytes. Mechanistically, *Ctcflos* operates at two regulatory levels, at transcriptional control of the brite characteristic gene program and as central regulator of alternative splicing during brite adipogenesis, exemplified by modulating transcription and isoform preference of *Prdm16*.

## Results

### Long noncoding RNAs are strongly regulated during brite adipocyte differentiation

To determine long noncoding RNAs of regulatory function in the browning process, we performed whole transcriptome analysis of undifferentiated and rosiglitazone-differentiated primary brite adipocytes derived from inguinal WAT (iWAT) of five mouse strains (SWR/J, BL/6J, A/J, AKR/J, and 129S6), as well as primary white adipocytes of 129S6 mice differentiated in the absence of rosiglitazone as described and validated previously (Data ref: Array Express E-MTAB-8344, 2019) (Data ref: Li *et al*, 2019b) (Fig 1A). This dataset allowed the comparison of the long noncoding RNA transcriptomes between (i) preadipocytes and differentiated brite cells, (ii) mouse strains of strong and weak browning propensity, and (iii) differentiated brite and white adipocytes.

Overall, we found that the identified long noncoding transcriptome, consisting of 5,741 transcribed lncRNAs (mean read number over all samples ≥ 3), clearly separated undifferentiated from differentiated adipocyte samples, demonstrating their transcriptional regulation during brite cell differentiation. Among them there might be lncRNAs that are functionally involved in the regulation of this process (Fig 1B and C). In principal component analysis and hierarchical clustering of all lncRNAs, preadipocyte samples clustered closely together apart from those of mature brite adipocytes. The long noncoding transcriptome thus displayed profound changes in the course of adipogenesis and browning and mirrored the state of differentiation. Furthermore, the distances between the strains clearly increased in the differentiated state, demonstrating that the

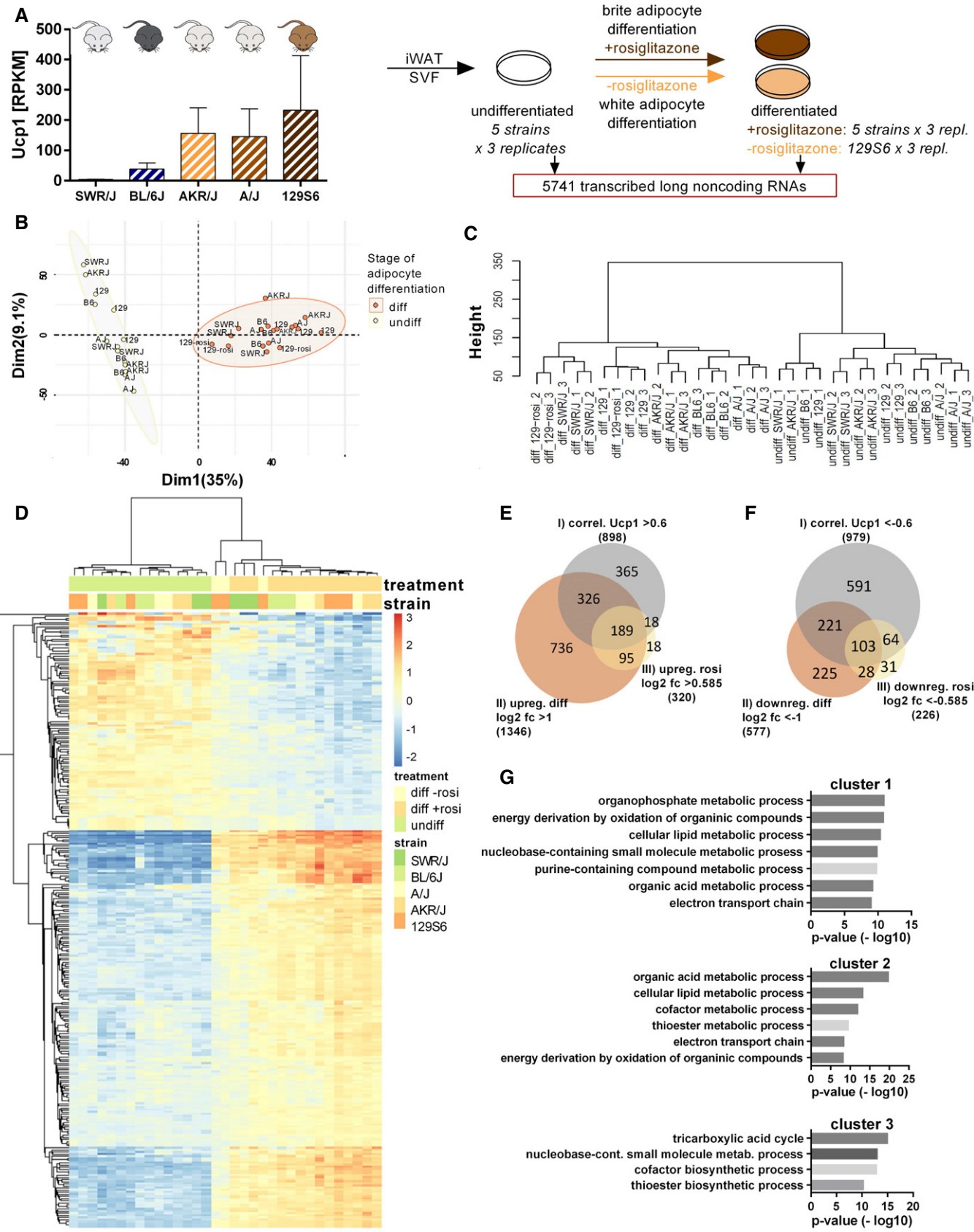

**Figure 1.**

**Figure 1.  Comparative transcriptome analysis reveals positive and negative lncRNA regulators in brite adipogenesis.**

A    Experimental design of transcriptome analysis. Primary preadipocytes and differentiated adipocytes ((w/o) rosiglitazone), isolated from inguinal white adipose tissue (iWAT) of five inbred mouse strains of varying browning propensity (evaluated based on Ucp1 transcript levels in RPKM (reads per kilobase per million mapped reads)). Mean values ± SD, $n = 3$ (biological replicates) (Data ref: Array Express E-MTAB-8344, 2019) (Data ref: Li *et al*, 2019b) were subjected to RNA sequencing to identify regulated long noncoding (lnc) RNAs.
B    Principal component analysis (PCA) of undifferentiated and differentiated adipocyte samples, $n = 3$ (biological replicates).
C    Hierarchical clustering of the long noncoding transcriptome of undifferentiated and differentiated adipocyte samples, $n = 3$ (biological replicates), Euclidean distance.
D    Heatmap, displaying row-wise centered log2 normalized counts.
E, F  Venn diagrams of positively and negatively regulated lncRNAs selected according to correlation with *Ucp1*, regulation during differentiation and regulation by rosiglitazone treatment, $n = 3$ (biological replicates).
G    GO term analysis of different mRNA clusters derived from lncRNA–mRNA coexpression network (according to Bai *et al*, 2017) that point toward lncRNA-associated cellular processes, hypergeometric test.

long noncoding transcriptome developed from a relatively common pattern to a strain-specific signature during the course of brite differentiation.

To reveal lncRNAs that are of functional relevance in brite adipogenesis, we set three selection criteria: We extracted lncRNAs that (i) correlate with *Ucp1* expression across the five mouse strains (Pearson $r > 0.6$ for positive correlations; Pearson $r < -0.6$ for negative correlations), (ii) are significantly regulated during differentiation of 129S6 adipocytes (log2 fold change (fc) (differentiated: undifferentiated) $> 1$; $< -1$ for up- and downregulation, respectively), and (iii) are affected by rosiglitazone (rosi) treatment in this strain (log2 fc (with rosi: without rosi) $> 0.585$; $< -0.585$ for induced and repressed lncRNAs, respectively). According to these three criteria, we identified 189 lncRNAs that positively correlated with *Ucp1* across the mouse strains, were upregulated during 129S6 brite adipogenesis, and induced by rosiglitazone (Fig 1D and E) (Dataset EV1), as well as 103 lncRNAs that negatively correlated with *Ucp1* and were downregulated during differentiation and rosiglitazone treatment (Fig 1D and F) (Dataset EV1) (Data ref: Array Express E-MTAB-8344, 2019). The proportion of positively regulated lncRNAs (3.3%) was similar to that of positively regulated protein-coding genes (3.7%) selected according to the same three criteria (Fig EV1A–C), proposing a similarly important role of this group of lncRNAs in the browning process.

In contrast to coding RNAs, it is difficult to generally annotate functions to lncRNAs due to their poor conservation (Bai *et al*, 2017). Nevertheless, functions of lncRNAs can be deduced from their correlation with mRNAs during dynamic physiological processes, with the hypothesis that overrepresented gene ontology (GO) terms for all mRNAs directly connected with at least one lncRNA within a correlation network cluster hint toward the biological functions of lncRNAs in this group (Cabili *et al*, 2011; Bai *et al*, 2017). To get an impression of the processes in which the identified, positively regulated lncRNAs might be functionally involved during brite adipogenesis, we generated a lncRNA–mRNA correlation network according to the above-described model. Clustering of the correlation network resulted in a subdivision into three main coexpression groups (clusters 1–3) (Fig 1G). Based on the presented GO terms of lncRNA-connected mRNAs, lncRNAs of cluster 1 and 2 were predicted to be involved in "electron transport chain" and "cellular lipid metabolism-" associated processes as well as "energy derivation by oxidation of organic compounds", while those of cluster 3 were associated with "tricarboxylic acid cycle" (Fig 1G). Together, our analyses suggest an appreciable contribution of lncRNAs to the regulation of brite adipogenesis and propose their involvement in brite cell energy metabolism and respiration.

### Long noncoding RNA *Ctcflos* stands out as the top candidate regulated in brite adipogenesis and brown fat thermogenesis

We further narrowed down the candidate lncRNAs to thermogenically active transcripts by testing their regulation in brown adipose tissue upon acute cold exposure *in vivo*. Only seven of the lncRNA transcripts, regulated in brite adipogenesis, were also significantly

**Figure 2.  *Ctcflos* stands out as strongest regulated lncRNA in brite adipogenesis *in vitro* and during iBAT activation *in vivo*.**

A    Venn diagram of lncRNAs positively regulated in brite adipogenesis *in vitro* (see Fig 1E) and induced in cold-mediated activation of interscapular brown adipose tissue (iBAT) *in vivo*. Seven lncRNAs are upregulated in both conditions.
B    Table of co-regulated lncRNAs. Three *Ctcflos* transcripts stand out as strongest regulated lncRNA candidates.
C–E  *Ctcflos* meets the three selection criteria for positively regulated lncRNAs in brite adipogenesis. (C) Correlation of *Ctcflos* (mean of transcript 1; 2; 3) with *Ucp1* transcript levels across differentiated brite adipocytes derived from the five mouse strains (transcript levels in RPKM). Mean values, $n = 3$ (biological replicates), Pearson correlation, *$P < 0.05$. (D) *Ctcflos* transcript levels (mean of transcript 1; 2; 3) in differentiated compared to undifferentiated brite adipocytes of 129S6 mice (transcript levels in RPKM). Mean and individual values, $n = 3$ (biological replicates), unpaired *t*-test, ****$P < 0.0001$. (E) *Ctcflos* transcript levels (mean of transcript 1; 2; 3) in differentiated brite (+rosi) compared to differentiated white (−rosi) adipocytes of 129S6 mice (transcript levels in RPKM). Mean and individual values, $n = 3$ (biological replicates), unpaired *t*-test, *$P < 0.05$.
F, G  Induction of *Ctcflos* along with *Ucp1 in vivo* in cold-activated iBAT (Data ref: Maurer *et al*, 2018). Time course of (F) *Ctcflos* and (G) *Ucp1* transcript levels in iBAT of 0, 6, 24, or 48 h cold (4°C) exposed C57BL/6J mice (transcript levels in RPKM). Mean and individual values, $n = 3$–4 (biological replicates), one-way ANOVA (Šídák-test), *$P < 0.05$, **$P < 0.01$, ***$P < 0.001$, ****$P < 0.0001$.
H, I  Induction of *Ctcflos* and *Ucp1 in vivo* during cold-induced iWAT browning. Relative expression levels of (H) *Ctcflos* transcript 1 and (I) *Ucp1* in iWAT of C57BL/6J mice held at 5°C for 1 week compared with age- and weight-matched mice held at 30°C or 23°C, assessed by quantitative PCR (qPCR). Mean and individual values, $n = 7$ (biological replicates), one-way ANOVA (Šídák-test), **$P < 0.01$, ****$P < 0.0001$.

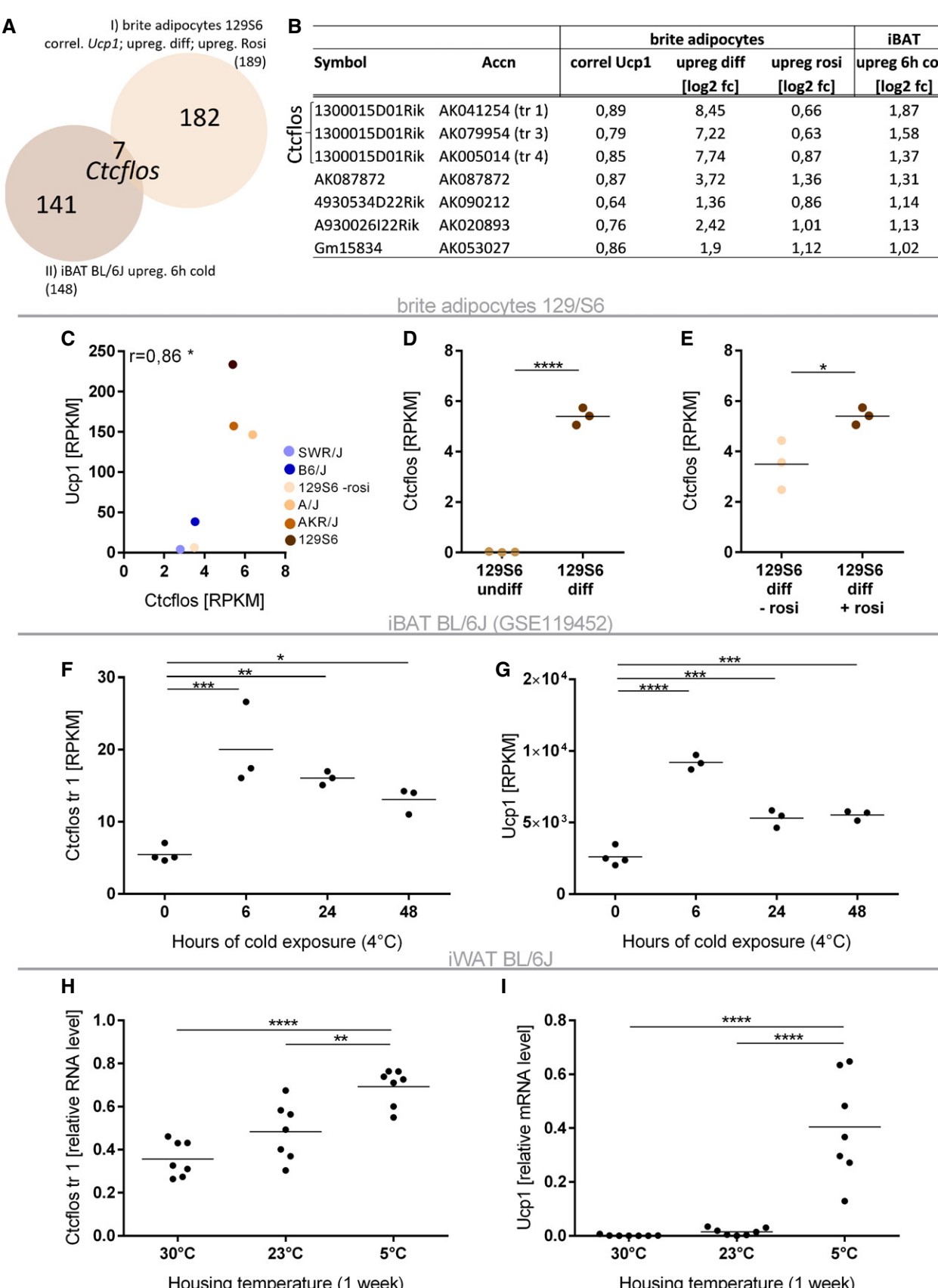

**Figure 2.**

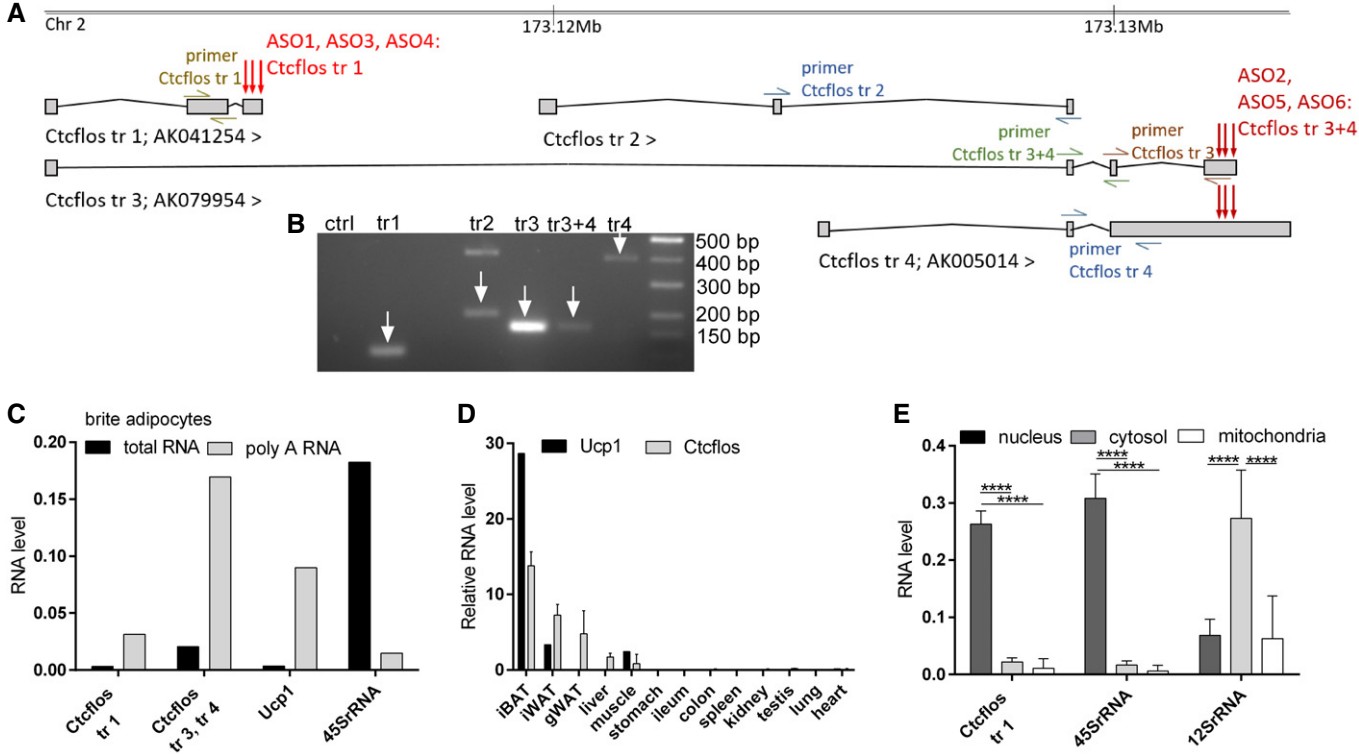

**Figure 3. *Ctcflos* is a polyadenylated, nuclear enriched, and adipose tissue predominantly expressed lncRNA.**

A   *Ctcflos* locus. Location and structure of annotated *Ctcflos* isoforms (Ensembl 99). Green, blue, and brown marks display primer-binding sites; red arrows mark target areas of LNA Gapmer antisense oligos.

B   Gel picture of *Ctcflos* transcripts expressed in differentiated brite adipocytes amplified by polymerase chain reaction (PCR) using transcript-specific primers.

C   Comparison of *Ctcflos*, *Ucp1*, and *45SrRNA* transcript abundance in total RNA and poly-A RNA fraction of primary differentiated brite adipocytes derived from 129S6 mice, assessed by qPCR, n = 1.

D   Relative transcript levels of *Ctcflos* and *Ucp1* in various tissues of 129S6 mice, assessed by qPCR. Mean values ± SD, n = 1 (biological replicate) for *Ucp1*, n = 3 (biological replicates) for *Ctcflos*.

E   *Ctcflos*, *45SrRNA*, and *12SrRNA* transcript abundance in nuclear, cytosolic, and mitochondrial subfractions of primary brite adipocytes of 129S6 mice, assessed by qPCR. Mean values ± SD, n = 6 (biological replicates), two-way ANOVA (Tukey test), ****$P < 0.0001$.

upregulated more than twofold in interscapular BAT (iBAT) of 6 h cold exposed BL/6J mice (Fig 2A and B) (Data ref: Gene Expression Omnibus GSE119452, 2018). The top three transcripts all constitute isoforms of lncRNA *Ctcflos*, on which we thus further focused for in-depth characterization.

LncRNA *Ctcflos* meets the selection criteria for the identification of putative regulatory lncRNAs as extracted from the transcriptome data and confirmed by qPCR. (i) Correlation analyses demonstrated that *Ctcflos* and *Ucp1* follow the same trend with higher transcript abundance in mouse strains of high browning capacity and lower abundance in those of minor browning propensity (Pearson $r = 0.86$) (Fig 2C). (ii) During differentiation of iWAT-derived precursor cells into mature brite adipocytes, *Ctcflos* expression was strongly elevated by 280- and 860-fold in RNA sequencing and qPCR data, respectively (Figs 2D and EV1D). (iii) *Ctcflos* transcript levels were enriched in brite compared with white cells (Figs 2E and EV1E). (iv) *In vivo*, *Ctcflos* expression levels were increased in brown fat in response to cold exposure of BL/6J mice (Fig 2F), which is correlated with *Ucp1* (Figs 2G and EV1F) (Data ref: Gene Expression Omnibus GSE119452, 2018). Similarly, cold-induced

iWAT browning (5°C, 1 week) (as indicated by increasing abundance of *Ucp1*-expressing adipocytes (Fig 2I)) in BL/6J mice was also accompanied by elevated *Ctcflos* gene expression compared with age- and weight-matched mice kept at room temperature and thermoneutrality (Fig 2H). Conclusively, our transcriptome analyses uncovered *Ctcflos* as the top regulated lncRNA candidate in thermogenic adipocyte development both *in vitro* and *in vivo*.

## *Ctcflos* is a polyadenylated, nuclear long noncoding RNA predominantly expressed in adipose tissues

*Ctcflos* gene (CCCTC-binding factor (zinc finger protein)-like, opposite strand; also 1300015D01Rik) is located on chromosome 2: (173,110,818–173,133,204) and transcribed into four major isoforms (*Ctcflos-tr1*; *Ctcflos-tr2*; *Ctcflos-tr3*; and *Ctcflos-tr4*) ranging from 469 to 3,476 bp length according to the Ensembl 99 annotation (Fig 3A and B). *Ctcflos* transcripts were predominantly present in the poly-A-enriched RNA fraction of primary brite adipocytes of 129S6 mice, indicating that they undergo further processing by polyadenylation (Figs 3C). *Ctcflos* was predominantly expressed in adipose tissues

including iBAT, iWAT, and gonadal WAT (gWAT) in descending order similar to *Ucp1* expression. Low transcript levels were observed in liver and muscle (Fig 3D). Concerning its subcellular localization, analyses of nuclear, cytosolic, and mitochondrial RNA fractions of primary brite adipocytes demonstrated a predominant retention of *Ctcflos* in the nucleus (Fig 3E). The lack of human orthologs of the four murine *Ctcflos* transcripts indicates that it is not conserved in humans.

To assess the protein-coding potential of *Ctcflos*, protein-coding potential scores were calculated using three different publicly available software tools: CPC2 (Kang *et al*, 2017), CPAT (Wang *et al*, 2013), and lncScore (Zhao *et al*, 2016). With a cutoff set at 0.4, all annotated *Ctcflos* transcripts were evaluated as non-protein-coding similar to *Neat1* a well-established adipose tissue expressed lncRNA, while *Ucp1*, as a positive control, was reliably evaluated as protein coding (Fig EV1G). For further validation, we subjected *Ctcflos* transcripts to *in silico* translation and blasted the translated amino acids against the RefSeq protein database. The lack of any significant hit supported that *Ctcflos* does not code for a protein (Fig EV1H).

### *Ctcflos* is essential for the browning process and thermogenic function of brite adipocytes

To evaluate the role of lncRNA *Ctcflos* in brite adipogenesis, we performed knockdown (KD) experiments using nuclear penetrating locked nucleic acid (LNA) Gapmer antisense oligonucleotides (ASO) targeting the three different *Ctcflos* isoforms that appeared as top candidates: ASO1, ASO3, and ASO4 for the knockdown of transcript 1 (ENSMUST00000144256.1; AK041254); ASO2, ASO5, and ASO6 for the knockdown of transcripts 3 and 4 (ENSMUST00000142207.7; AK079954 and ENSMUST00000153523.1; AK005014) (Fig 3A). A third ASO without cellular target served as negative control. We favored the use of LNA Gapmer ASOs over RNAi, since they achieve more efficient suppression of nuclear located lncRNAs such as *Ctcflos* (Lennox & Behlke, 2016). The presence of locked nucleic acids at the ASO flanks furthermore enables high target-binding affinity and specificity. Based on the time course expression pattern of *Ctcflos* over proliferation, induction, and differentiation, where *Ctcflos* transcript levels first significantly increased very early during differentiation (day 1), followed by a second boost in the later phase (day 5) (Fig 4A), we chose an early knockdown time point (KD on day 1 and analysis on day 4 of differentiation) and a late knockdown time point (KD on day 5 and analysis on day 8 of differentiation) in order to investigate the effects of *Ctcflos* deficiency during the onset of brite adipogenesis and in mature brite cells (Figs 4B and EV2A). *Ctcflos* transcripts were efficiently reduced to around 20 and 30% in early and late knockdown, respectively (Figs 4C and D, and EV2C and D). Early loss of *Ctcflos* lncRNA induced a profound reduction in *Ucp1* mRNA, the key marker of brite and brown adipocytes, to 23, 27, 39 and 51, 19, 21% by ASO1, ASO3, ASO4 and ASO2, ASO5, ASO6, respectively (Fig 4E). Similar KD effects were observed for pre-mature *Ucp1* transcripts in the nucleus, indicating that reduced *Ucp1* mRNA evolves mainly from impaired *Ucp1* gene transcription rather than mRNA processing or nuclear export (Fig EV1I). The impairment of brite adipogenesis was further confirmed at protein level as demonstrated by diminished UCP1 signals in Western blot and immunofluorescent analyses (Fig 4G–I). Brite

and brown marker genes *Cidea* and *Cox7a1* were also notably lower transcribed (Fig 4F). Knockdown of *Ctcflos* tended to shift cell morphology toward reduced brite characteristic multilocularity, with slightly lower lipid droplet numbers (Fig EV1J). The observed trend for increased lipid droplet sizes was not significant (Fig EV1K–M). *Ctcflos*-deficient cells maintained the ability to differentiate into lipid-loaded adipocytes, as indicated by the lack of significant reduction in Oil red O staining (Fig EV1N and O). *Ctcflos* KD thus interferes specifically with the brite determining gene program while only marginally affecting adipogenesis *per se*.

Loss of the brite characteristic gene expression profile further translated into severe disturbances of the thermogenic potential of *Ctcflos*-deficient cells, as reflected in profoundly shifted respiration profiles. *Ctcflos*-deficient cells possessed lower levels of basal oxygen consumption and a profound reduction in isoproterenol-stimulated, UCP1-dependent uncoupled respiration (Fig 4J–L). The prominent difference in isoproterenol-stimulated respiration between control and knockdowns was present in wild type while absent in *Ucp1* knockout cells validating the specificity of the requirement of *Ctcflos* in UCP1-dependent uncoupled respiration (Fig 4M–O). UCP1-mediated thermogenesis thereby depends not only on the expression level of UCP1 but also the extent of UCP1 activation by lipolytically released free fatty acids. Since lipolysis was not reduced in isoproterenol-stimulated *Ctcflos*-deficient cells (Fig 4P), it is the strong reduction in UCP1 expression that was primarily responsible for the impaired thermogenic capacity. Decreased basal oxygen consumption was caused by mildly reduced mitochondrial biogenesis and thus impaired oxidative capacity of mitochondria as demonstrated by diminished Mito Tracker signals (Figs 4R and S, and EV1P) and reduced abundance of COX4 protein (Fig 4Q). Notably, the KD-induced reduction in UCP1-dependent uncoupled respiration was stronger than that of basal oxygen consumption. Impaired isoproterenol-induced respiration was thus not merely the consequence of reduced mitochondrial content but was mainly caused by the decrease in UCP1. *Ctcflos* deficiency during early phase of brite differentiation thus severely impaired brite adipocyte formation and function.

Consistent with these strong effects of *Ctcflos* deficiency during early differentiation, KD of *Ctcflos* in proliferating preadipocytes had a similar impact on *Ucp1* expression (Fig EV2A and B). Also in mature adipocytes, the *Ctcflos* KD effect remained, albeit slightly alleviated. *Ctcflos* deficiency generated during late phase of differentiation, when *Ctcflos* is highest expressed, accordingly reduced brite marker gene expression and uncoupled respiration (Fig EV2A and C–J). *Ctcflos* is thus also required for maintaining the brite phenotype in mature adipocytes.

Moreover, *Ctcflos* is likewise essential for differentiation and function of brown adipocytes. *Ctcflos* KD at the onset of differentiation clearly affected *Ucp1* transcription and UCP1-mediated uncoupled thermogenesis in brown adipocytes (Fig EV2K–O). In white adipocytes, *Ctcflos* deficiency had only minor effects (Fig EV2P), demonstrating its specific requirement in thermogenic adipocyte development.

Together, our data validate *Ctcflos* as an essential component in the specific control of thermogenic programming in brite and brown adipocytes. It is required for the acquisition of transcriptional and functional characteristics in developing brite adipocytes and for the maintenance of the thermogenic phenotype in mature cells.

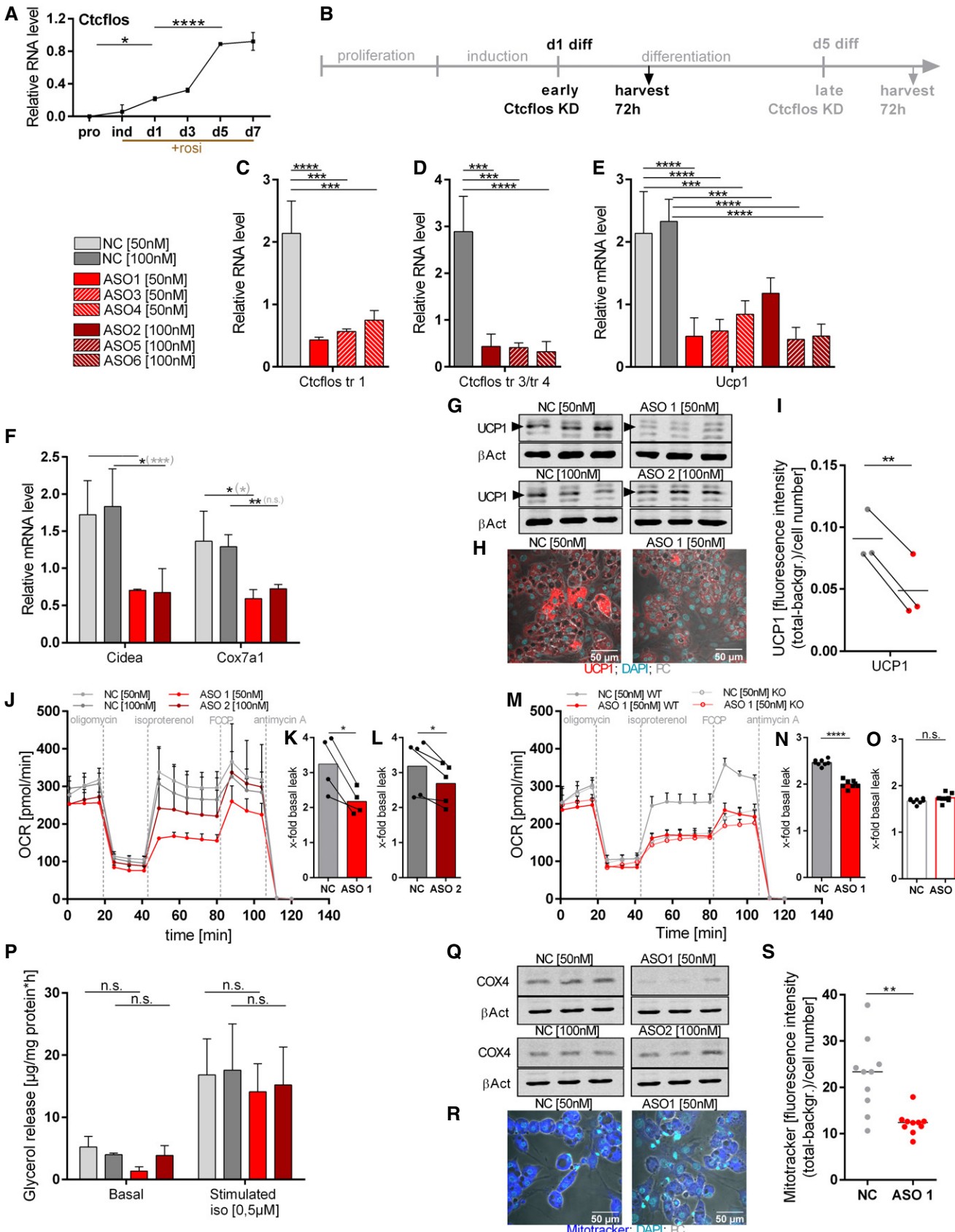

Figure 4.

**Figure 4.** ***Ctcflos* is essential for brite adipocyte differentiation and thermogenic function.**

A   Time course of relative *Ctcflos* expression across proliferation, induction, and differentiation of primary brite adipocytes, assessed by qPCR. One representative curve of three experiments with comparative results, mean values ± SD of 3 technical replicates, one-way ANOVA, *P < 0.05, ****P < 0.0001.

B   Design of *Ctcflos* knockdown (KD) experiments. *Ctcflos* KD was performed at the first day after induction, and cells were harvested and analyzed 72 h later.

C, D   Efficiency of *Ctcflos* KD by (C) LNA Gapmer ASO1, ASO3, and ASO4 targeting *Ctcflos* transcript 1 and (D) LNA Gapmer ASO2, ASO5, and ASO6 targeting *Ctcflos* transcripts 3 and 4 compared with nontargeting controls, assessed by qPCR. Mean values ± SD, n = 3 (biological replicates), one-way ANOVA (Šídák-test ), ***P < 0.001, ****P < 0.0001.

E   Relative *Ucp1* transcript levels in response to *Ctcflos* tr1, *Ctcflos* tr3 and *Ctcflos* tr4 KD by ASO 1, 3, 4 and 2, 5 6, respectively, compared to nontargeting controls, assessed by qPCR. Mean values ± SD, n = 3–4 (biological replicates), one-way ANOVA (Šídák-test), ***P < 0.001, ****P < 0.0001.

F   Relative expression levels of *cell death-inducing DNA fragmentation factor alpha-like effector A* (*Cidea*) and *cytochrome c oxidase subunit 7a1* (*Cox7a1*) in *Ctcflos* tr1, 3 and 4 KD compared with control samples, assessed by qPCR. Mean values ± SD, n = 3 (biological replicates)-4, unpaired *t*-tests or in gray parenthesis two-way ANOVA (Šídák-test), n.s. P > 0.05, *P < 0.05, **P < 0.01, ***P < 0.001.

G   Western blot analysis of UCP1 protein abundance in response to *Ctcflos* tr1, *Ctcflos* tr3, and *Ctcflos* tr4 KD by ASO 1 and 2, respectively, compared with nontargeting controls. Actin-β as loading control. Arrow heads mark UCP1-specific bands.

H   Immunocytochemistry of UCP1 protein in response to *Ctcflos* tr1 KD by ASO 1 (UCP1 (red); DNA staining by DAPI (green); phase contrast (PC) (gray)).

I   Quantification of UCP1 fluorescent signal as whole image fluorescence intensity subtracted by background and normalized to cell number. Mean and individual values, n = 3 (biological replicates) (with 7–8 analyzed images for each biological replicate), paired *t*-test **P < 0.01.

J–O   Effect of *Ctcflos* KD on brite adipocyte respiratory capacity. Time course of oxygen consumption rates in primary brite adipocytes 72 h after treatment (day 1 diff), comparing (J–L) *Ctcflos* tr1 KD (ASO 1) and *Ctcflos* tr3 and *Ctcflos* tr4 KD (ASO 2) with their respective controls and (M-O) *Ctcflos* tr1 KD (ASO 1) and its control in UCP1 wild-type (WT) and UCP1 knockout (KO) cells, measured by microplate-based respirometry (Seahorse XF96 Analyzer). (M) Oxygen consumption is recorded under basal conditions and in response to successive injection of oligomycin (5 μM), isoproterenol (1.5 μM), FCCP (1 μM), and antimycin A (5 μM) to determine basal leak, UCP1-dependent uncoupled, maximal and non-mitochondrial respiration, respectively. Data are expressed after deduction of non-mitochondrial respiration. (J) Mean values ± SD, n = 4–5 (biological replicates). (M) Mean value ± SD, n = 7 (technical replicates). (K), (L), (N), (O) Quantification of isoproterenol-stimulated UCP1-mediated uncoupled respiration, expressed as fold of basal leak respiration. Mean and single values, (K) n = 4 (biological replicates), paired *t*-test *P < 0.05, (L) n = 5 (biological replicates), paired *t*-test *P < 0.05, (N, O) n = 7 (technical replicates), unpaired *t*-test, n.s. P > 0.05, ****P < 0.0001.

P   Effect of *Ctcflos* KD on brite adipocyte lipolysis. Basal and isoproterenol (iso) (0.5 μM) stimulated glycerol release comparing *Ctcflos* tr1 (ASO 1), tr3 and tr4 (ASO 2) KD and their controls. Mean values ± SD, n = 4 (biological replicates), two-way ANOVA (Šídák-test), n.s. P > 0.05.

Q–S   Impact of *Ctcflos* KD on mitochondrial biogenesis. (Q) Western blot analysis of cytochrome C oxidase subunit 4 (COX4) protein level in *Ctcflos* tr1 (ASO 1), tr3, and tr4 (ASO 2) KD and controls. Actin-β as loading control. (R) Microscopic images of Mito Tracker-stained brite adipocytes (Mito tracker (blue); DNA staining by DAPI (green); phase contrast (PC) (gray)) and (S) quantification of Mito Tracker staining as whole image fluorescence intensity subtracted by background and normalized to cell number comparing *Ctcflos* tr1 (ASO 1) KD and control. Mean and individual values, n = 10 (technical replicates) the experiment was repeated a second time, for a total of two biological replicates, for the second replication, see (Fig EV1P) (with 10 analyzed images), unpaired *t*-test **P < 0.01.

Source data are available online for this figure.

## Global transcriptome analysis reveals *Ctcflos*-dependent thermogenic programming mediated by transcriptional regulation of *Prdm16*

Across the presented experiments, ASO 1-mediated KD of *Ctcflos* tr1 had a stronger impact than ASO 2-induced *Ctcflos* tr3 and *Ctcflos* tr4 deficiency. This suggests distinct mechanistic functionalities of the *Ctcflos* transcripts and attributes Ctcflos tr1 the dominant role in brite adipogenesis. Our further global transcriptome investigations therefore concentrated on Ctcflos tr1.

Previous reports demonstrated that lncRNAs act *in cis*, through transcriptional regulation of their neighboring genes (Ding *et al*, 2018). We therefore investigated the effect of *Ctcflos* KD on the transcription of its up- and downstream neighbors *Ctcfl* and *Pck1*. *Ctcfl* was very low transcribed and nonsignificantly changed in *Ctcflos* KD adipocytes in qPCR analysis (Fig EV2Q), while *Pck1* was downregulated in qPCR analysis of *Ctcflos* tr1 deficiency (Fig EV2T). Neither knockdown of *Ctcfl* nor knockdown of *Pck1* did affect *Ucp1* gene transcription (Fig EV2R, S, U and V), demonstrating that *Ctcflos* functions in brite adipogenesis independently of its neighbor genes *Ctcfl* and *Pck1*. This suggests that *Ctcflos* acts *in trans*, demanding for comprehensive analysis of *Ctcflos* KD-induced transcriptional changes.

To gain insight into the molecular basis of *Ctcflos* function during brite adipogenesis, we analyzed transcriptome changes in brite adipocytes upon *Ctcflos* KD both acutely (after 24 h) and after 72 h of adaptation (Fig 5A). Shortly after KD, we identified 1,437 differentially expressed transcripts (P adj.< 0.05 DESeq2) with 780 upregulated and 657 downregulated transcripts, while after 72 h, 692 transcripts

were enriched and 715 were reduced comparing KD vs. control cells (Fig 5B and C). Our global transcriptome analysis reliably confirmed the observed *Ctcflos* KD phenotype with clearly reduced expression of *Ucp1* along with other thermogenic marker genes (Fig 5D) and broad downregulation of several subunits of respiratory chain complexes (Fig EV3A and B) after 72 h of adaptation. Consistently, GO term analysis of downregulated genes reveals a strong overrepresentation of several terms associated with mitochondria, respiratory chain, thermogenesis, and brown fat cell differentiation (Fig 5E). The expression of general adipogenesis markers, in contrast, is hardly affected, again supporting the specificity of *Ctcflos* action in thermogenic gene programming rather than general adipogenesis (Fig EV3C). Notably, among the regulated genes after KD we found several key factors of the transcription regulatory network of brite and brown adipogenesis such as *Prdm16, Nuclear factor I/A* (*Nfia*), *Early B cell factor 1* (*Ebf1*), and *Ppar$\gamma$* (Fig 5F). They were transcriptionally reduced directly (24 h) after the knockdown representing potential proximal regulatory modules of *Ctcflos*. These early changes subsequently (after 72 h) converged in reduced transcription of *Ucp1* gene regulatory element-binding factors and coregulators including *thyroid hormone receptor $\beta$* (*Thr$\beta$*), *Ppar$\alpha$*, *Cebp$\beta$*, and *Pgc1$\alpha$* which can finally mediate reduced *Ucp1* transcription (Fig 5F). We thus demonstrate that lncRNA *Ctcflos* is required for early formation of the core transcription regulatory network that programs brite adipocyte development. Beyond this, transcriptome analysis in mature brite KD cells similarly confirmed *Ctcflos* requirement for brite cell marker gene expression but dispensability for general adipogenic gene transcription (Fig EV3D and E).

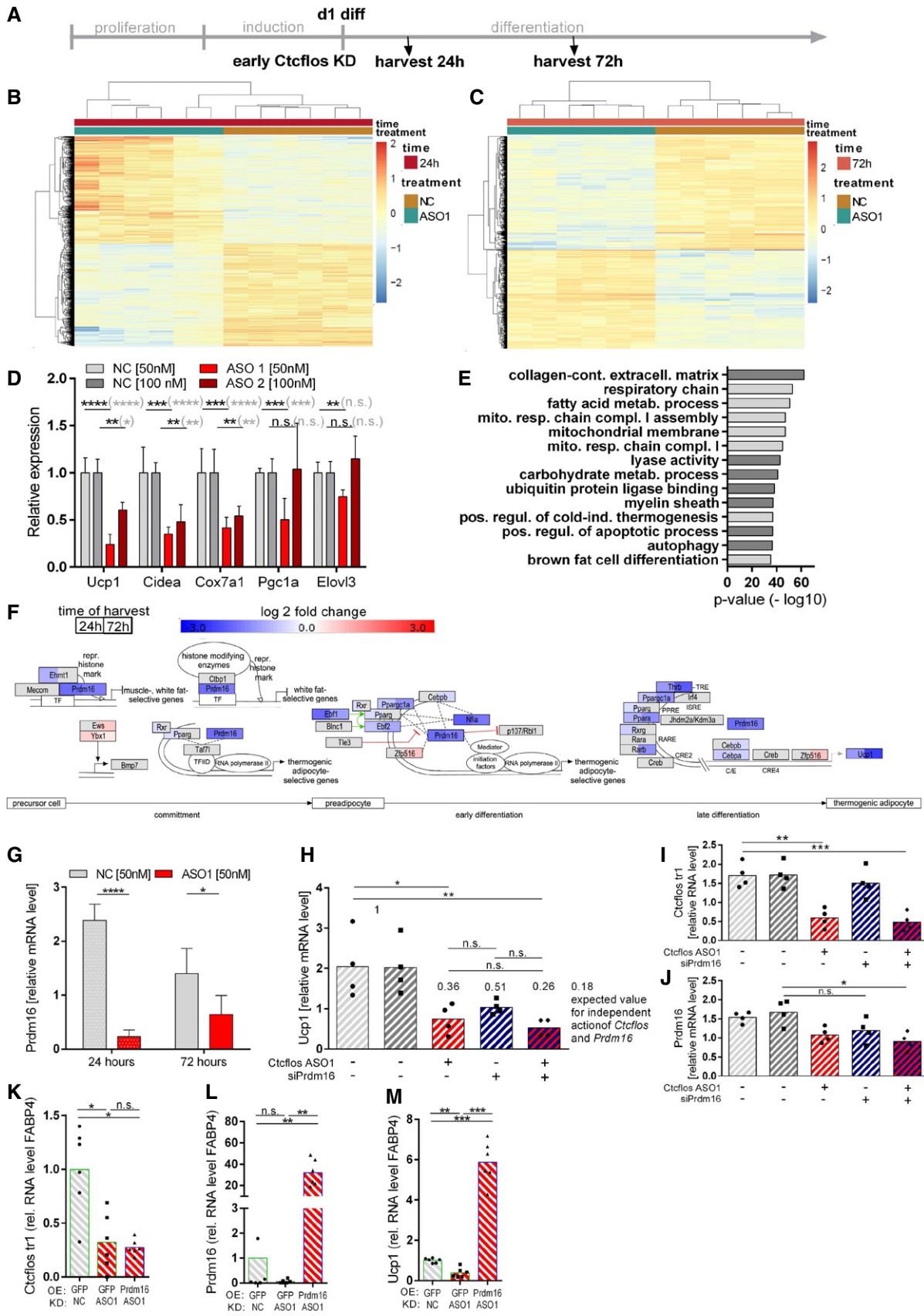

Figure 5.

**Figure 5.  Global transcriptome profiling reveals *Ctcflos*-dependent thermogenic regulatory program.**

A       Experimental setup of two-time point transcriptome analysis. *Ctcflos* tr1 (ASO 1), tr3, and tr4 (ASO 2) or control treatments were performed at the first day after induction (d1 diff), and cells were harvested for transcriptome analysis 24 and 72 h later.

B, C    Heat maps of differentially expressed genes between *Ctcflos* tr1 (ASO 1) KD and control cells (B) 24 h after the KD (780 up- and 657 downregulated genes) and (C) 72 h after the KD (692 up- and 715 downregulated genes). Row-wise centered log2 normalized counts are displayed.

D       Relative expression levels of brite adipocyte marker genes *uncoupling protein 1* (*Ucp1*), *cell death-inducing DNA fragmentation factor alpha-like effector A* (*Cidea*), *cytochrome c oxidase subunit 7a1* (*Cox7a1*), *peroxisome proliferator-activated receptor γ coactivator 1α* (*Pgc1a*), and *elongation of very long chain fatty acid-like 3* (*Elovl3*), comparing *Ctcflos* tr1 (ASO 1) and tr3 and tr4 (ASO 2) KD with their respective controls. Mean values ± SD, $n = 6$ (biological replicates), unpaired *t*-tests or in gray parenthesis two-way ANOVA (Šídák-test) n.s. $P > 0.05$, **$P < 0.01$, ***$P < 0.001$, ****$P < 0.0001$.

E       GO term analysis of genes downregulated 72 h after the knockdown. Negative log10 *P*-values of overrepresented GO terms, hypergeometric test.

F       Transcriptomic effect of *Ctcflos* tr1 (ASO 1) KD on brown and brite core regulatory gene expression network comparing *Ctcflos* tr1 (ASO 1) KD with controls. Visualized by Path Visio software. Color code visualizes log2 fold change of genes 24 h (left part of box) and 72 h (right part of box) after the KD. Red t-bars represent inhibitory effects; green arrows represent activating effects.

G–J     Impact of *Ctcflos* KD on *Ucp1* expression is largely mediated via reduced *Prdm16* transcription. (G) Relative transcript levels of *PR domain containing 16* (*Prdm16*) 24 and 72 h after the knockdown of *Ctcflos* tr1 (ASO1) compared to control cells, assessed by qPCR, Mean values ± SD, $n = 3$ (biological replicates), two-way ANOVA (Šídák-test), ****$P < 0.0001$, *$P < 0.05$. (H) Relative *Ucp1*, (I) *Ctcflos* tr1 and (J) *Prdm16* transcript levels in single and double KD (day 1 diff) of *Ctcflos* tr1 (ASO 1) and *Prdm16* (siPrdm16), nontargeting LNA Gapmer (light gray) and siRNA (dark gray) controls, assessed by qPCR, mean and individual values, one-way ANOVA (Šídák-test), n.s. $P > 0.05$, *$P < 0.05$, **$P < 0.01$, ***$P < 0.001$.

K–M     Rescue of *Ctcflos* KD impact on *Ucp1* gene transcription by *Prdm16* overexpression. Primary iWAT cells infected with Prdm16- or turbo green fluorescent protein (GFP)-expressing viral particles at the second day of proliferation, followed by reverse transfection at the first day of differentiation using nontargeting control (NC) or Ctcflos-targeting ASO1. Gene expression analyzed 72 h later by qPCR. (K) Transcript levels of *Ctcflos* tr1 relative to Fabp4. (L) Transcript levels of Prdm16 relative to Fabp4. (M) Transcript levels of Ucp1 relative to Fabp4. Mean and individual values. Each graph presents pooled data from two experiments slightly of varying virus titers. RM one-way ANOVA (Tukey test), n.s. $P > 0.05$, *$P < 0.05$, **$P < 0.01$, ***$P < 0.001$.

Similar to *Ctcflos*, PRDM16 functions across the entire process of brite adipocyte development, including commitment of preadipocytes, induction of differentiation, and maintenance of thermogenic gene expression (Ishibashi & Seale, 2015). With its fast impact on *Prdm16* transcription, *Ctcflos* therefore affects a well-established early key regulator in thermogenic fat cells that is essential and sufficient for the recruitment of brite adipocytes in WAT (Seale *et al*, 2011). The observed *Ctcflos* KD phenotype thus likely occurs in part as consequence of impaired *Prdm16* expression. In line with this hypothesis, other components of the thermogenic regulatory network including *Pparα*, *C/ebpβ*, *Ebf2*, and *Pgc1α*, which are downstream targets of *Prdm16*, were also downregulated by *Ctcflos* KD in chronological succession to impaired *Prdm16* transcription (Fig 5F). To provide further evidence that *Prdm16* functions as a proximal component in *Ctcflos*-dependent signaling during brite adipogenesis, we first experimentally confirmed its regulation by *Ctcflos* through qPCR analysis of *Ctcflos* KD cells (Fig 5G). Through genetic interaction assay via pairwise KD of *Ctcflos* and *Prdm16*, we further found that the double KD effect on the suppression of *Ucp1* expression was 1.4-fold less severe than the theoretically calculated KD effect (multiplicative model of single KD effects) that would be expected if *Ctcflos* would work completely independent of *Prdm16* (Fig 5H–J). *Ucp1* transcript levels were similar comparing *Ctcflos* single KD with *Ctcflos*-*Prdm16* double KD (at a significance level of α = 0.1 in a paired TOST equivalence test) proposing that the double KD did not result in a considerable additive effect (Fig EV3F). The assumed epistasis between *Ctcflos* and *Prdm16* was additionally tested in rescue experiments. For this, we conducted *Ctcflos* KD in *Prdm16* overexpressing cells and compared the resulting phenotype with that of non-*Prdm16*-overexpressing control and *Ctcflos* KD cells. *Prdm16* overexpression in *Ctcflos* deficiency could rescue *Ucp1* gene expression supporting that *Prdm16* acts as a downstream mediator of *Ctcflos* in brite adipogenesis (Figs 5K–M and EV3G–I). A considerable portion of the *Ctcflos* KD effect might thus be attributed to the regulation of *Prdm16* transcription though not excluding other effectors and pathways of *Ctcflos* function in the browning process.

### *Ctcflos* modulates *Prdm16* isoform preference toward its thermogenic gene program-promoting short isoform

In addition to the central regulatory function of *Ctcflos* in the brite transcription program, the deeper investigation of transcriptome changes in response to *Ctcflos* KD revealed immediate effects on the splicing machinery. For the early KD of *Ctcflos* (24 h), our gene ontology (GO) term analysis of upregulated genes showed strong overrepresentation of several GO terms related to molecular functions of the splicing machinery (Fig 6A). This enrichment of molecular functions related to splicing appeared to be a proximal consequence of *Ctcflos* deficiency, as it was stronger expressed in direct response to the KD (Fig 6A). In a first explorative experiment, we found that inhibition of the splicing process by isogink-getin, a general splicing inhibitor, largely attenuated thermogenic differentiation while adipogenesis was less impaired, as judged by a decrease in *Ucp1* per *Fabp4* mRNA levels (Fig 6B–D). Alternative splicing, a mechanism generating different transcripts from a single gene, can convey alternative functions and thus affect phenotypic plasticity. We observed transcriptional upregulation of several components of the spliceosome and splicing factors shortly (24 h) after the depletion of *Ctcflos*, pointing toward changes in the splicing machinery (Fig EV4A). Stronger immunofluorescent signals of serine/arginine-rich splicing factor and nuclear speckle marker SC35 (SRSF2) confirmed this (Figs 6E and F, and EV4B and C) and further revealed an increased number of nuclear speckles per nucleus in direct response to *Ctcflos* KD (Figs 6G, and EV4D and E). As speckles serve as subnuclear storage compartments for components of the splicing machinery and sites of splicing factor activation (Spector & Lamond, 2011), this further underpinned the modulation of splicing activity in direct response to the KD. A negative correlation between SC35 and UCP1 protein abundance across individual *Ctcflos* KD cells that is absent in control cells additionally supported a causal relationship between modulations in splicing and reduced UCP1 expression (Figs 6H and I, and EV4F–Q).

It has been shown previously that lncRNAs are important players in nuclear speckle function and modulation of alternative splicing (Tripathi *et al*, 2010; Cooper *et al*, 2014). Moreover, alternative transcript isoforms with divergent roles in thermogenic adipocyte differentiation have been identified (Lin, 2015). Most interestingly, two main protein-coding isoforms of *Prdm16* are described that vary in the utilization of exon 16 and different stop codons within exon 17, with the short isoform exhibiting more-prominent influence on brown fat development (Fig 6J) (Chi & Lin, 2018). Thus, not only the total PRDM16 transcript abundance but also its relative isoform composition influences thermogenic adipogenesis. To check the potential influence of the observed splicing machinery changes on alternative splicing of *Prdm16*, we determined the relative abundance of both isoforms during brite adipogenesis and in response to *Ctcflos* KD. Similar to previous observations in brown adipocytes, we here detected a shift in the relative expression of the two isoforms from equal to short-form dominant expression during brite adipocyte differentiation (Fig 6L). In *Ctcflos* deficiency, however, this shift in alternative splicing was blunted (Fig 6K and M). The difference in abundance between the two isoforms was considerably reduced. The splicing pattern of *Prdm16* in *Ctcflos* KD cells thus clearly deviated from that of thermogenic adipocytes and resembled to that of preadipocytes. qPCR confirmed that total *Prdm16* expression decreased predominantly at the expense of *Prdm16 short*

isoform while *Prdm16 long* isoform was not significantly reduced (Fig 6O–Q and T–V). This occurred already 24 h after the KD (Fig 6O–S) and further remained (Fig 6T–X), thus preceding and accompanying the manifestation of the impaired browning phenotype. The mean fold change (control/*Ctcflos* KD) of *Prdm16 short* was significantly higher than the fold change (control/Ctcflos KD) of *Prdm16 long* isoform (for ASO1 at 24 and 72 h, for ASO2 at 72 h after KD) (Fig 6R, S, W and X). The prominent role of *Prdm16* short isoform in driving brite adipogenic programming was further supported by its positive correlation with *Ucp1* transcript levels that occurred in control but was lost in KD samples (Fig 6N). Therefore, *Ctcflos* not only regulates the overall abundance of *Prdm16*, but also modulates its relative isoform composition, which further fine-tunes its function toward promoting thermogenic adipogenesis. Together, we demonstrate that *Ctcflos* acts at two mechanistic levels, controlling both the quantity and the quality of the brite master regulator *Prdm16*, which further establishes *Ctcflos* as a lncRNA regulator of alternative splicing.

### *Ctcflos* acts as a core regulator of alternative splicing in brite adipogenesis

Based on the above insights, we aimed to systematically evaluate the role of *Ctcflos* in driving alternative splicing during brite

**Figure 6. *Ctcflos* is a core regulator of alternative splicing in brite adipogenesis promoting the expression of thermogenic *Prdm16* isoform.**

A    GO term analysis of genes upregulated in response to *Ctcflos* knockdown. In dark gray, negative log10 *P*-values of overrepresented GO terms 24 h after KD and percentages of genes belonging to the respective GO terms among upregulated and among all genes. Small numbers next to the bars show rank positions of GO terms. In light gray, corresponding values for the same GO terms after 72 h, hypergeometric test.

B–D    Impact of splicing inhibition by different concentrations of general splicing inhibitor isoginkgetin on brite adipogenesis, (B) expressed as *Ucp1* relative to *Fabp4* mRNA levels, (C) *Ucp1* relative to *Gtf2b* and (D) *Fabp4* relative to *Gtf2b*. Mean values ± SD, *n* = 3 (biological replicates), one-way ANOVA (Šídák-test), n.s. *P* > 0.05, *\*P* < 0.05, *\*\*P* < 0.01, *\*\*\*P* < 0.001, *\*\*\*\*P* < 0.0001.

E–I    Effect of *Ctcflos* KD on serine/arginine-rich splicing factor 2 (SC35/SRSF2) and nuclear speckle abundance in *Ctcflos* tr1 (ASO 1) KD and control cells (KD day 1 of differentiation, analysis after 48h). (E) Immunocytochemistry of SC35. SC35 signal (blue) alone (middle images) and SC35 (blue), DNA staining by DAPI (green) and phase contrast (PC) (gray) combined (outer images). (F) Quantification of SC35 fluorescence signal as whole image fluorescence intensity subtracted by background and normalized to total cell number. Mean and individual values, one representative replicate of three experiments with comparative results (see Fig EV4B and C), unpaired *t*-test, *\*\*P* < 0.01. (G) Quantification of nuclear speckle number as mean number of SC35 signals per nucleus. Mean and individual values, one representative replicate of three experiments with comparative results (see Fig EV4D and E), unpaired *t*-test, *\*\*P* < 0.01. (H, I) Correlation of SC35 and UCP1 fluorescence signals across individual cells in (H) control and (I) *Ctcflos* tr1 (ASO 1) KD cells. Extreme values were excluded to avoid distortion of correlation analysis. One representative replicate of four experiments with comparable results (see Fig EV4F-Q). Pearson correlation, *\*\*\*\*P* < 0.0001.

J–X    Influence of *Ctcflos* KD on *Prdm16* alternative splicing. (J) Splice graph of *Prdm16* and resulting *Prdm16* long and short isoforms. Primers used to amplify both *Prdm16* isoforms at once by PCR are displayed in orange; primers used to amplify specifically Prdm16 short and Prdm16 long, respectively, are displayed in blue. (K) Gel picture of PCR products using exon 16 spanning primers to detect *Prdm16* long (upper band (348 bp)) and *Prdm16* short (lower band (173 bp)) comparing *Ctcflos* tr1 (ASO 1) KD and control samples. (L) Time course of relative expression of *Prdm16* total, long and short isoforms during brite adipogenesis, assessed by qPCR using isoform-specific primers. Mean values ± SD, one representative replicate of two experiments with comparable results, two-way ANOVA (Šídák-test), *\*\*\*P* < 0.001, *\*\*\*\*P* < 0.0001. (M) Levels of *Prdm16* long and *short* isoforms in *Ctcflos* tr1 (ASO1) KD and control cells. Signal intensities quantified from PCR gel picture subtracted by background and normalized to overall mean signal intensity of the respective biological replicate. Mean and individual values. Two-way ANOVA (Šídák-test), *\*P* < 0.05, *\*\*\*\*P* < 0.0001. (N) Correlation of *Prdm16* short and *Ucp1* expression levels across *Ctcflos* tr1 (ASO1) KD and control samples and within both groups. Pearson correlation, n.s. *P* > 0.05, *\*\*P* < 0.01, *\*\*\*\*P* < 0.0001.(O) Relative *Prdm16* total, (P) relative *Prdm16* long and (Q) relative *Prdm16* short transcript levels 24 h after *Ctcflos* KD, comparing *Ctcflos* tr1 (ASO1) and tr3, tr4 (ASO2) KD and control samples, assessed by qPCR, Mean values ± SD, *n* = 3–4 (biological replicates), unpaired *t*-tests, n.s. *P* > 0.05, *\*P* < 0.05, *\*\*P* < 0.01, *\*\*\*P* < 0.001. (R, S) Fold change of *Prdm16* short and *Prdm16* long (R) ASO1 to NC and (S) ASO2 to NC 24 h after *Ctcflos* KD, *n* = 3–4 (biological replicates), unpaired *t*-tests, n.s. *P* > 0.05, *\*\*P* < 0.01. (T) Relative *Prdm16* total, (U) relative *Prdm16* long and (V) relative *Prdm16* short transcript levels 72 h after *Ctcflos* KD, comparing *Ctcflos* tr1 (ASO 1) KD and control cells, assessed by qPCR, mean values ± SD, *n* = 3–4 (biological replicates), unpaired *t*-tests, *\*P* < 0.05, *\*\*\*P* < 0.001. (W, X) Fold change of *Prdm16* short and *Prdm16* long (W) ASO1 to NC and (X) ASO2 to NC 72 h after *Ctcflos* KD, *n* = 3–4 (biological replicates), unpaired *t*-tests, *\*P* < 0.05.

Y, Z    Global transcriptome analysis of *Ctcflos* KD impact on alternative splicing (KD day 1 of differentiation, analysis after 24 h). (Y) MA-plot of differential alternative splicing analysis using SGSeq algorithm comparing *Ctcflos* tr1 (ASO 1) KD with controls. Each data point depicts a splice event. Splice events of differential abundance between *Ctcflos* tr1 (ASO 1) KD and controls (*P*-adj. < 0.05) are colored in red. (Z) Overview of selected genes involved in adipogenesis, lipid metabolism, mitochondria, or splicing, with differential splicing profiles comparing *Ctcflos* tr1 (ASO 1) KD and control samples. Genes previously described to be spliced into alternative isoforms of varying function in brown adipogenesis are marked in red.

AA    Schematic summary of *Ctcflos* mechanistic functions in brite adipogenesis.

Source data are available online for this figure.

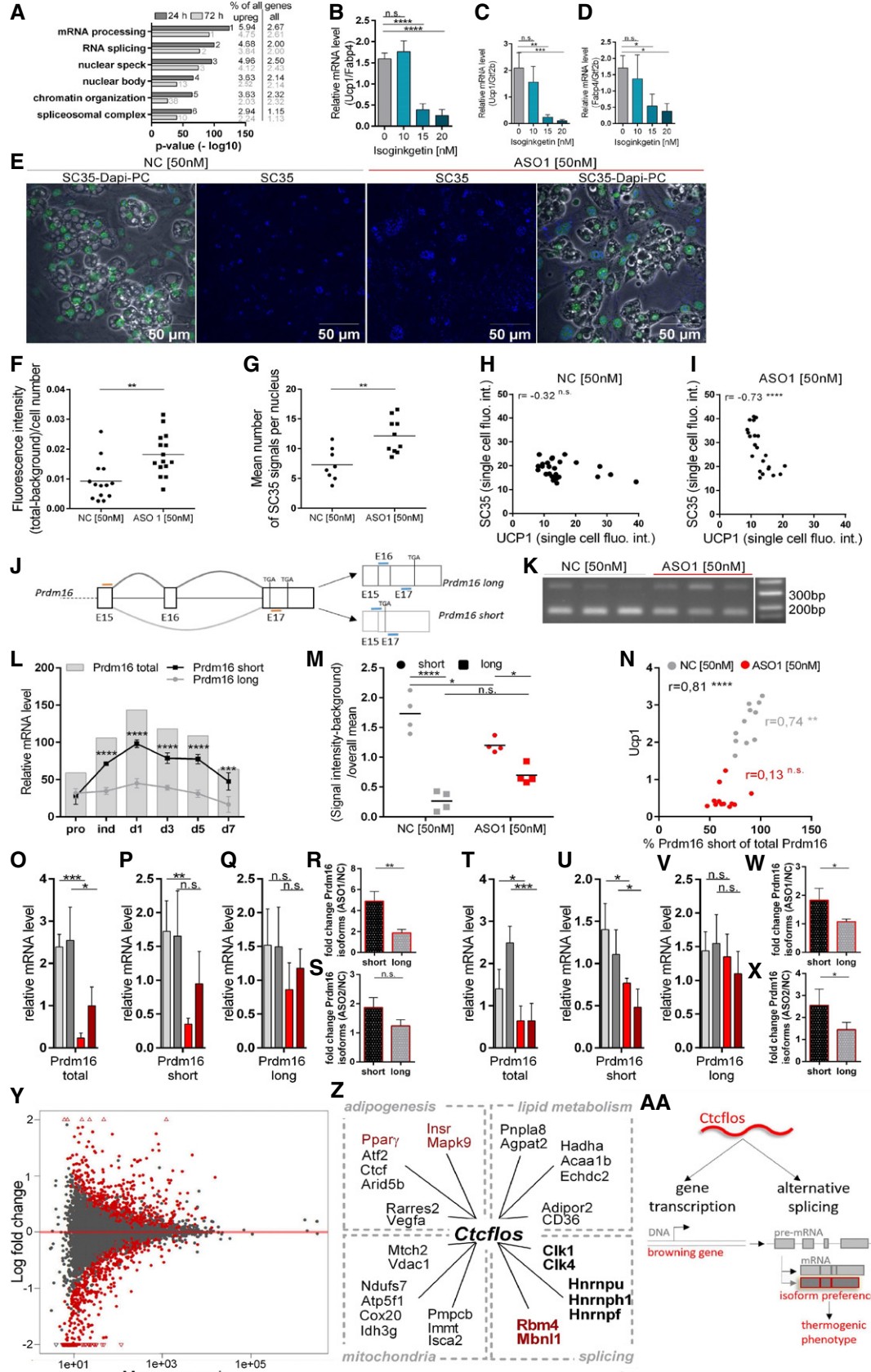

**Figure 6.**

adipogenesis. To this end, we performed 150 bp paired-end deep RNA sequencing comparing *Ctcflos* KD with control adipocytes. Identifying differential splice events with SGSeq analysis (Goldstein *et al,* 2016) revealed as much as 455 transcripts that were targeted for alternative splicing by *Ctcflos* (Fig 6Y). Notably, some of the differentially spliced genes have been characterized to be spliced into alternative transcripts of different functions in adipogenesis, including *Insulin receptor* (*Insr*), *Ppar$\gamma$*, *Mitogen-activated protein kinase 9* (*Mapk9*), *Muscleblind-like splicing factor 1* (*Mbnl1*), and *RNA-binding motif protein 4* (*Rbm4*) (Entingh *et al,* 2003; Cooper *et al,* 2014; Lin *et al,* 2014; Vernia *et al,* 2016; Hung & Lin, 2019). *Insr* pre-mRNA for instance can be processed into exon 11 (ENSMUSE00001381444; 36bp) including (Insr$_{+ex11}$) or excluding (Insr$_{-ex11}$) isoforms, of which Insr$_{+ex11}$ exerts the prominent role in p38-MAPK signaling and brown adipogenesis (Lin *et al,* 2014). Consistently, our splicing analysis suggests lower exon 11 inclusion in *Ctcflos* KD cells (Fig EV5A). Confirming the validity and reliability of our analysis, we observed a lower ratio of Insr$_{+ex11}$ to Insr$_{-ex11}$ in *Ctcflos* deficiency 72 h after the KD through RT–PCR gel analysis (Fig EV5B–E). The shift from similar expression of both isoforms (24 h post-treatment) to predominance of Insr$_{+ex11}$ (72 h post-treatment) that occurred during brite adipocyte differentiation (Fig EV5F and G data for NC) was less pronounced in the absence of *Ctcflos* (Fig EV5F and G data for ASO1). Interestingly, among the identified differentially spliced genes, regulated by *Ctcflos*, RNA-binding motif protein 4 (RBM4) has been described as a crucial regulator of alternative splicing and gene transcription during brown fat development (Lin *et al,* 2014). It constitutes a regulatory network in programming splicing profiles of several genes (including *Insr*, *Ppar$\gamma$*, *Mbnl1*, *Preadipocyte factor 1* (*Pref-1*), and *Prdm16*) that express transcripts of varying roles in brown adipogenesis (Fig EV5H) (Lin *et al,* 2014; Chi & Lin, 2018; Hung & Lin, 2019). By modulating *Rbm4*, *Ctcflos* thus acts upstream of a complex splicing regulatory network that fosters the thermogenic gene program (Fig EV5H). Supporting this hypothesis, *Ctcflos* KD evoked a major change in alternative splicing patterns of many genes involved in adipogenesis, lipid metabolism, mitochondrial function, and splicing, beyond the above-described splice events (Fig 6Z). Roughly one third of the differentially spliced genes were identified as candidates for browning regulation in our previous study (Li *et al,* 2019a).

Taken together, lncRNA *Ctcflos* acts as a core regulator of alternative splicing, coordinating a complex splicing network to induce major changes in alternative splicing patterns of many genes in brite adipogenesis. Together with its role in transcription regulation, this attributes *Ctcflos* a complex regulatory potential (Fig 6AA). The ability to modulate two steps within the gene expression process of multiple browning-associated genes, exemplified by transcriptional regulation and post-transcriptional alternative splicing of *Prdm16*, confers *Ctcflos* an indispensable role in brite adipocyte development and function.

## Discussion

Understanding the intricate regulatory mechanisms that govern recruitment and activity of thermogenic brite adipocytes constitutes the fundamental prerequisite for their deployment as therapeutic strategy against obesity and associated comorbidities. Brite adipogenesis is a multilayered process that requires coordinated integration of multiple regulatory factors to build up a brite characteristic proteome composition and function. In this context, lncRNAs are versatile regulators that can orchestrate cellular differentiation by acting at several levels of gene expression simultaneously. Here, we employed the naturally diverging cell-intrinsic propensity of iWAT browning between inbred mouse strains revealing 189 lncRNAs of potential involvement in the regulation of brite adipogenesis and thermogenesis. Among them, we characterized nuclear lncRNA *Ctcflos* as a critical, versatile component in the brite regulatory network that controls thermogenic gene programing both at transcriptional and post-transcriptional levels. As such, *Ctcflos* is required for transcriptional induction and isoform preference of key brite adipogenesis regulator *Prdm16* along with several other browning-associated factors, making it essential for functional brite adipocyte development. Consistent with this pivotal role in the thermogenic gene program, *Ctcflos* likewise presents a significant component in classical brown adipogenesis.

Our deepened knowledge on the human genome and transcriptome gained by the achievements of the Human Genome and the ENCODE projects taught us that our canonical view on cellular differentiation and function needs to be multiplexed by an additional layer of regulation by noncoding RNAs. The broad expansion in the application of whole transcriptome profiling confirmed the indispensable functional involvement of lncRNAs in a multitude of biological processes including white and brown adipogenesis. *SRA1*, *lnc-RAP-1*, *lnc-leptin*/ *lncOb*, and *Plnc1* are reported regulators of white adipogenesis, while *Blnc1*, *lnc-BATE-1/10*, and *H19* were identified to be essential for brown adipocyte differentiation and function (Xu *et al,* 2010; Sun *et al,* 2013; Zhao *et al,* 2014; Alvarez-Dominguez *et al,* 2015; Ding *et al,* 2018; Lo *et al,* 2018; Schmidt *et al,* 2018; Dallner *et al,* 2019; Zhu *et al,* 2019). These findings highlight a role for lncRNAs in adipose biology and thermogenesis. In line, our model of naturally varying browning propensities proposes a considerable contribution of the long noncoding transcriptome to differentiation of brite adipocytes. It discloses a number of potent lncRNA candidates that are worth further investigation, providing a rich resource for exploiting the role of lncRNAs in brite adipogenesis.

Among this group of lncRNAs, *Ctcflos* stood out as the top candidate by its strong regulation during brite adipogenesis *in vitro* and its cold-induced upregulation in iBAT and iWAT *in vivo* as validated in different mouse strains (129S6 and BL/6J). Validating our own data, *Ctcflos* transcripts 3 and 4 are also upregulated in published transcriptome analysis in iBAT of cold exposed mice (Marcher *et al* 2015). Independent datasets thus suggest a potential regulatory role of *Ctcflos* in thermogenic gene programming. Consistently, our study demonstrates that *Ctcflos* plays an essential and specific role in thermogenic adipocyte development and function. Depletion studies reveal its requisite contribution to brite and brown adipogenesis, but vast dispensability in general adipogenesis and white adipocyte differentiation. Strongest impacts of *Ctcflos* occur during the early phase of differentiation, emphasizing its particular contribution to the early switch toward the thermogenic program. This temporally coincides with critical transcription time windows of brite master regulator *Prdm16* and brite cell key functional entity *Ucp1* that both reach their maximum levels within the first half of the differentiation process. Eliminating *Ctcflos* during this early phase in the thermogenic differentiation program induces severe impairments of brite and brown adipocyte development as evidenced on

transcriptional and functional levels. Typical brite cell characteristics such as multilocularity, mitochondrial biogenesis, UCP1 expression, and thermogenic uncoupled respiration are strongly impaired due to early loss of *Ctcflos*. Besides the KD of *Ctcflos*, it would be valuable to rescue the observed phenotype by overexpression of a mutated version of *Ctcflos* that cannot be targeted by the ASOs but remains its functionality in brite adipogenesis. This experimental approach, however, requires exact annotation of *Ctcflos* transcript sequences and the identification of functional ribonucleotide motifs crucial for *Ctcflos* activity that need to be clarified in future studies.

The functional importance of lncRNAs in thermogenic adipocyte differentiation is well in line with the versatile nature of lncRNA molecules, which can coordinate gene expression at several levels through different mechanistic themes. Accordingly, the multilayered process of brite adipogenesis demands delicate spatial and temporal orchestration of epigenetic, transcriptional, post-transcriptional, and translational regulation to build up a brite cell-specific protein machinery that specifies cell morphology and function. Generally, transcriptional control of brite genes largely determines the composition of the cellular machinery, while cell type characteristic post-transcriptional modification such as alternative splicing allows further fine-tuning of its functionality (Pan *et al*, 2008; Wang *et al*, 2008; Nilsen & Graveley, 2010; Yang *et al*, 2016). Around 95% of human genes can be spliced into distinct isoforms that can convey different or even opposite functioning (Pan *et al*, 2008). The modulation of isoform profiles thus allows precise adaptation of the cells equipment to meet the changing requirements during cellular differentiation (or can confer cell type-specific action to broadly expressed factors) (Lin *et al*, 2014; Singer *et al*, 2019). Accordingly, it has been shown previously that coordination of alternative splicing plays a significant role in adipocyte differentiation. Brown adipogenesis and white fat expansion involve broad changes in RNA splicing profiles (Vernia *et al*, 2016; Chi & Lin, 2018). Several key adipogenic factors with functionally varying alternative isoforms as well as adipocyte-related splicing factors have been identified (Lin, 2015). Hence, although often neglected, modulation of post-transcriptional processing is a further pivotal driver of cell type-specific differentiation, along with transcriptional and translational regulations.

In this context, it is of great interest that lncRNAs act as versatile regulators able to modulate many steps within the gene expression process including chromatin modification, enhancer activity, transcription, pre-mRNA splicing, miRNA processing, nuclear export, translation, and protein stability (Geisler & Coller, 2013; Marchese *et al*, 2017). In fact, it is a peculiarity of lncRNAs that the same lncRNA can act at multiple levels simultaneously, which strongly increases their potential to coordinate the regulatory interplay during cellular differentiation.

Consistent with this notion, our study revealed a complex functionality for *Ctcflos*. Our data disclose that *Ctcflos* acts as a central regulator of both gene transcription (1) and alternative splicing (2), conferring it an effective regulatory potential in brite adipogenesis.

1   Concerning gene transcription regulation, our data demonstrated that *Ctcflos* is required to build up the core thermogenic gene transcription program in differentiating adipocytes. Applying a two-time point strategy, our transcriptome analysis thereby goes beyond a single snapshot and provides a chronological view on *Ctcflos* KD-induced changes by disentangling direct effects from consecutive and adaptive changes. In *Ctcflos* deficiency, several key factors of the thermogenic gene expression machinery (including *Prdm16*, *Nfia*, *Ebf1*, and *Pparγ*) are transcriptionally downregulated in direct response to the KD. This in turn is responsible for the compromised expression of brown fat selective genes and *Ucp1*, forming the basis for the observed phenotype of impaired mitochondrial biogenesis and diminished UCP1-dependent thermogenic respiratory capacity that manifests in chronological succession. Through which exact mechanism, however, *Ctcflos* exerts its transcription regulatory function is unclear and warrants further study.

2   Concerning *Ctcflos*-dependent modulation of alternative splicing, we observed in parallel to the transcriptional alterations, broad changes in the splicing machinery and increased numbers of nuclear speckles, the sites for storage and activation of splicing factors, predicting altered splicing behavior in *Ctcflos* KD cells. Deep global analysis of splicing profiles correspondingly revealed that *Ctcflos* exerts broad influence on alternative splicing in developing brite adipocytes within 24 h, largely affecting browning-associated genes. Among them there are genes identified before to possess isoforms of varying impact on adipogenesis, such as *Insr*, *Pparγ*, *Mapk9*, *Mbnl1*, and *Prdm16* (evaluated by PCR and qPCR) (Cooper *et al*, 2014; Lin *et al*, 2014; Vernia *et al*, 2016; Chi & Lin, 2018; Hung & Lin, 2019). Along with the transcriptional changes, these early deviations from brite characteristic splicing profiles can contribute to the consecutive phenotype. Moreover, splicing-associated factors with impact on adipogenesis are included in the *Ctcflos*-dependent differentially spliced transcripts. Among these factors, *hnRNPU* has been shown to interact with lncRNAs and to be required for brown adipogenesis; *Rbm4* is a pivotal alternative splicing modulator during brown fat cell differentiation, and *Clk1* is required for splicing factor (SRp40) phosphorylation during adipogenesis (Cooper *et al*, 2014; Lin *et al*, 2014; Alvarez-Dominguez *et al*, 2015; Mi *et al*, 2017; Sun *et al*, 2017). Modulating the splicing machinery itself confers *Ctcflos* the potential to control alternative splicing on a broad scale and qualifies it as a crucial player in this process. Similarly, other lncRNAs, such as *ASCO, LINC01133, Malat1, and Enod40,* that act as splicing modulators had been shown before (Romero-Barrios *et al*, 2018). By modulating the isoform profiles of a multitude of browning-associated genes and fine-tuning the splicing machinery, *Ctcflos* could thus further drive brite adipogenesis, complementing its transcription regulatory impact.

Notably, both *Ctcflos* dependently modulated processes, transcription and alternative splicing, occur temporally in parallel directly after *Ctcflos* KD, proposing a coregulation. In fact, transcription and splicing are tightly associated procedures, being coordinately regulated in the nucleus, both spatially and temporally. Transcription factors and RNA polymerase II were previously described to directly or indirectly (through cofactors) interact with splicing factors to concomitantly modulate the transcription rate and splicing profile of a target gene (Cramer *et al*, 1999; Monsalve *et al*, 2000; Auboeuf *et al*, 2004; Das *et al*, 2007). Interestingly, nuclear speckles function as storage, assembly, and modification compartments of transcription- and splicing-associated factors.

Phosphorylation and dephosphorylation cycles of splicing factors, transcription factors, and cofactors within the speckles coordinate their activity, assembly to higher order complexes and translocation to the site of transcription and RNA processing, attributing nuclear speckles a pivotal role in the coordination of transcription and splicing. The fact that nuclear speckles are increased in number in *Ctcflos* deficiency provokes the hypothesis that *Ctcflos* is involved in their functionality. The increased nuclear speckle signal might point toward impaired activation and translocation of splicing- and transcription-associated factors out of the speckles in *Ctcflos* deficiency. Interestingly, lncRNA *Malat1* has been shown to reside within nuclear speckles of HeLa cells and to modulate local phosphorylation and dephosphorylation cycles of splicing factors, demonstrating the regulatory capacity of lncRNAs in these events (Tripathi *et al*, 2010). In this context, an interaction of *Ctcflos* with transcription factors, coregulators, splicing factors or kinases, and proteases within nuclear speckles to modulate their phosphorylation and thus activity states, their translocation or complex assembly could be envisioned but still awaits experimental exploration. To examine such a putative role of *Ctcflos* in nuclear speckle function, future analysis should investigate the subnuclear location of *Ctcflos*, phosphorylation states of speckle-associated splicing and transcription factors as well as direct interaction partners of *Ctcflos*.

The functionality of *Ctcflos* in transcription and post-transcriptional alternative splicing is exemplified in the brite master regulator *Prdm16*. PRDM16 is well established as a cell-autonomous early key mediator of brown and brite adipocyte identity (Seale *et al*, 2011). It directs cell fate into the thermogenic adipocyte lineage and co-activates several transcription factors to boost brown and brite cell characteristic gene expression (Ishibashi & Seale, 2015; Kajimura, 2015; Chi & Cohen, 2016). Chronological transcriptome analysis of direct and adaptive *Ctcflos* KD effects demonstrates that *Ctcflos* induces *Prdm16* gene expression and is required for consecutive *Prdm16*-dependent brite gene programming. Combined *Ctcflos* and *Prdm16* KD further supported *Prdm16* as proximal mediator of *Ctcflos* activity in brite adipogenesis. Rescue of *Prdm16* expression in *Ctcflos* KD adipocytes, abolishing the impairment of *Ucp1* gene expression, further substantiated the involvement of *Prdm16* in *Ctcflos*-dependent brite adipogenesis. Additionally, *Ctcflos* directs the alternative splicing of *Prdm16* into its short isoform, which gains predominance during the course of brite adipogenesis and is required for full activation of essential thermogenic genes (Chi & Lin, 2018). *Ctcflos* thus controls both *Prdm16* transcript abundance (quantity) and its isoform-dependent functional specification (quality) to drive thermogenic adipogenesis. This finding illustrates how lncRNAs can modulate transcription factor activities to achieve cell- and developmental stage-specific regulation.

Conclusively, our comparative transcriptome analyses of diverging iWAT browning propensities among inbred mouse strains uncovered adipose tissue enriched lncRNA *Ctcflos* that orchestrates early gene programming toward brite and brown adipogenesis. It coordinates two levels of gene expression by regulating transcription and alternative splicing of multiple browning-associated genes, particularly the master regulator *Prdm16*, to induce the brite-selective gene program and to fine-tune the functionality of the brite adipogenic machinery toward thermogenesis. With its comprehensive impact on splicing of brite adipocyte factors and the splicing machinery itself, *Ctcflos* works as a central coordinator of brite adipogenesis-promoting alternative splicing. As a versatile regulator,

*Ctcflos* accomplishes an essential role in brite and brown fat cell differentiation, mitochondrial biogenesis, and thermogenesis. Our study emphasizes the multidimensional nature of thermogenic adipogenesis and underlines the coordinative regulatory potential of lncRNAs in this process.

# Material and Methods

### Animals

Male 129S6sv/ev, A/J, AKR/J, SWR/J, C57BL6/J, and 129S1/SvImJ mice (UCP1-KO mice and wild-type littermates) were bred at the animal facility of Technical University of Munich and kept at room temperature (23°C ± 1°C), 55% relative humidity, 12-h:12-h light:dark cycle and ad libitum access to water and food (regular chow diet V1124-3M-Z; ssniff Spezialdiäten GmbH, Germany). At the age of 5–6 weeks, they were used for whole organ extraction or preparation of primary cell cultures of interscapular brown or inguinal white and brite adipocytes. Previously conducted animal studies that provided data or material for this manuscript were performed according to the German Animal Welfare based on approved licenses.

### Primary cell isolation and culture

The stromal vascular fractions of inguinal and interscapular fat pads were isolated, grown, and differentiated into white, brite, or brown adipocytes as previously described (Li *et al*, 2014; Oeckl *et al*, 2020). In brief, dissected fat pads where digested with collagenase, stromal vascular fraction cells were seeded (6-well, 12-well or 24-well plates), grown to confluence in adipocyte culture medium (DMEM, heat-inactivated FBS (20%), penicillin/streptomycin (0.4%), gentamycin (0.4%), fungizone (0.2%)) and induced for 48 h in the presence of insulin (5 µg/ml), 3,3′,5-triiodo-l-thyronine (T3) (1 nM), isobutylmethylxanthine (IBMX) (0.5 mM ), indomethacin (125 µM), dexamethasone (1 µM), and rosiglitazone (1 µM) (for brite and brown adipocytes) in culture medium with 10% FBS. Differentiation was continued with insulin (5 µg/ml), T3 (1 nM), and rosiglitazone (1 µM) (for brite and brown adipocytes) in culture medium with 10% FBS for 3–7 days. Medium was changed every other day.

### RNA sequencing

For the identification of lncRNAs of importance in iWAT browning, we analyzed the RNA-seq data set on undifferentiated and differentiated (w/o rosiglitazone) primary adipocytes of five inbred mouse strains (C57BL/6J, 129S6sv/ev, A/J, AKR/J, and SWR/J) that was published previously (Data ref: Li *et al*, 2019b) and that is available at (https://www.ebi.ac.uk/arrayexpress/) accession number E-MTAB-8344. For the identification of lncRNAs of potential functional relevance in brite adipogenesis, sequenced tags were mapped against the GRCm38/mm10 genome assembly, and uniquely identified hits were assigned to transcripts of the ElDorado database, a Genomatix genome annotation, based on publicly available reference assemblies, including Refseq, Ensembl, and GenBank (by scanning sequenced tags for the shortest unique sequences (SUS) defining a specific position in the target genome). LncRNAs were extracted from alignments against the GenBank and analyzed for

differential expression using the Genomatix DESeq2 pipeline. Transcript coverage rates were assigned in reads per kilobase per million mapped reads (RPKM). Correlation with Ucp1 > 0.6 or < −0.6, significant regulation during differentiation with log2 fold change > 1 or < −1, and significant regulation in response to rosiglitazone treatment with log2 fold change > 0.585 or < −0.585 were set as criteria to identify brite adipogenesis relevant lncRNAs. For heatmap generation, transcript-level count data were normalized using the regularized logarithm transformation, subset to those lncRNAs identified as being of functional relevance as described above, row-mean centered and clustered using hierarchical clustering.

For transcriptome wide analysis of *Ctcflos* KD effects, RNA samples of KD and control cells, harvested directly (24 h) or later (72 h) after the KD at day 1 or day 5 of differentiation, were subjected to bulk RNA barcoding and sequencing (BRB-seq) as described (Li *et al*, 2019a). All samples were tested for RNA integrity using RNA 6000 Nano chip (Agilent, Santa Clara, CA, USA) and reverse transcribed by SuperScriptTM II Reverse Transcriptase (Lifetech 18064014) using specific barcoded oligo-dT primers. Samples pools were purified by DNA clean and concentrator kit (Zymo Research #D4014), followed by exonuclease I treatment (New England Biolabs #M0293S) and second strand synthesis by nick translation (Alpern *et al*, 2019) to generate double-stranded (ds) cDNA. cDNA was purified with AMPure magnetic beads (30 µl, 0.6×) (Beckman Coulter, #A63881) and eluted in water (20 µl). Libraries were prepared by tagmentation of full-length ds cDNA (5 ng) with Tn5 enzyme (1 µl, 11 µM) and amplified (15 cycles). AMPure beads (Beckman Coulter, #A63881) were used to size-select fragments of 200–1,000 bp in two rounds (0.5× beads and 0.7× beads, respectively). Libraries were profiled with the High Sensitivity NGS Fragment Analysis Kit (Advanced Analytical, #DNF-474), measured with the Qubit dsDNA HS Assay Kit (Invitrogen, #Q32851), pooled, and sequenced on the Illumina NextSeq 500 platform with custom primer and High Output v2 kit (75 cycles) (Illumina, #FC-404-2005). Sequenced tags were aligned to the NCBI Reference Sequence Database, and differential expression analysis was performed with DESeq2 using Genomatix Software Suite.

For the analysis of *Ctcflos* KD-dependent changes in splicing patterns, *Ctcflos* KD and control samples were deep sequenced using Illumina HiSeq 4000 platform (Helmholtz Center Munich). Libraries of six samples were pooled into one sequencing lane and sequenced in 150 bp paired-end reads, ~40 mio reads per sample. Reads were aligned to the GRCm38 genome assembly by STAR_2.5.3a. The obtained BAM files were used to construct splice graphs based on Gencode vM21 exon annotation using SGSeq algorithm (Goldstein *et al*, 2016). Splice events detected based on splice junction spanning reads were further tested for differential occurrence using DEXSeq (Anders *et al*, 2012).

## Principal component and hierarchical clustering analysis

In principal component and hierarchical clustering analysis, expression data of all lncRNAs with a mean read number larger than three across all five mouse strains (undifferentiated and differentiated) were used. Principal component analysis was performed with prcomp R function (with centering and scaling) and factoextra R package. For hierarchical clustering, the distance matrix of the lncRNA expression data was computed and hierarchical clustering

was performed with dist (with scaling) and hclust R functions, respectively.

## Coding potential determination

Protein-coding potential scores were calculated with publicly available software, CPC2 (Kang *et al*, 2017), CPAT (Wang *et al*, 2013), and lncScore (Zhao *et al*, 2016). CPC2 was run using default parameters. LncScore was run using the hexamer and training data distributed with the software. CPAT was run via the web interface. Cutoff for protein-coding probability was set at 0.4. In silico translation and blasting of transcripts against the RefSeq protein database were performed using blastx with restriction to the plus strand.

## lncRNA–mRNA coexpression network analysis

Gene network analysis was done as previously described (Bai *et al*, 2017) Briefly, transcript-level count data were normalized using the regularized logarithm transformation and subset to those transcripts identified as being of functional relevance (by correlation with Ucp1, regulation during differentiation, and rosiglitazone treatment) as described above. Partial correlations and FDR were calculated for all pairs of transcripts using the GeneNet R package. Partial correlations with FDR < 5% were used to create a graph, where vertices represent transcripts and edges significant correlations. Using the fast greedy modularity optimization algorithm (igraph R package), networks of commonly regulated transcripts were determined. mRNAs without a direct link to a lncRNA were removed from the analysis and the remaining mRNAs used for GO analysis using a parent–child algorithm (topGO R package).

## Gene ontology

Overrepresentation analyses of gene ontology terms within gene sets were conducted by free InCroMAP software (Wrzodek *et al*, 2013).

## Knockdown by antisense locked nucleic acid gapmers and siRNAs

Knockdown experiments of lncRNA and mRNA transcripts were performed using antisense locked nucleic acid (LNA) Gapmer antisense oligonucleotides (ASOs) (Qiagen), short interference RNAs (siRNAs) (Eurofins), or dicer substrate short interference RNAs (dsiRNAs) (IDT), respectively, as previously described (Li *et al*, 2017). Briefly, primary cells were trypsinised at confluence, after induction or at day 5 of differentiation and reverse transfected into 12-well, 24-well cell culture, or XF96 Seahorse culture plates containing 25 min pre-incubations of transfection reagent (Lipofectamine RNAiMAX (5 µl/ml) (Life Technologies)) and LNA Gapmers or siRNA (50 or 100 nM) in differentiation medium without antibiotics. After 24 h, cells were either directly harvested (direct knockdown effects) or treated with fresh differentiation medium and harvested/analyzed after another 48 h (consecutive knockdown effects). Nontargeting LNA Gapmers and siRNAs were used as controls. Sequences of LNA Gapmer ASOs and siRNAs are listed below:

LNA Gapmer ASO 1 targeting Ctcflos tr 1: 5′-GCTTGGGTGG AGGATT-3′.

LNA Gapmer ASO 2 targeting Ctcflos tr 3 and 4: 5′-ATTTGA CAACCACTG-3′.

LNA Gapmer ASO 3 targeting Ctcflos tr 1: 5′-GCTTGGGT GGAGGATT-3′.

LNA Gapmer ASO 4 targeting Ctcflos tr 1: 5′-CTGGCTTGGGT GGAGG-3′.

LNA Gapmer ASO 5 targeting Ctcflos tr 3 and 4: 5′-AGTCAG ACACCTTTTA-3′.

LNA Gapmer ASO 6 targeting Ctcflos tr 3 and 4: 5′-AGACTA ATTGCTAGCG-3′.

LNA Gapmer Negative control A: 5′-AACACGTCTATACGC −3′.

siRNA Pck1: 5′-GGAAGTTCGTGGAAGGCAA-3′.

dsiRNA Ctcfl: 5′-GAATTCTAAAGTCAACCTGAGAATC-3′.

siRNA Prdm16: 5′-GAAGAGCGTGAGTACAAATTT-3′.

## Lentiviral overexpression of Prdm16

Lentiviral overexpression of *Prdm16* was performed in primary proliferating preadipocytes derived from iWAT followed by *Ctcflos* KD at the first day of differentiation. FLAG-Prdm16 open reading frame was PCR-amplified from MSCV-Prdm16 (gift from Bruce Spiegelman (Addgene plasmid # 15504; http://n2t.net/addgene: 15504; RRID:Addgene_15504) (Seale *et al*, 2007)) and cloned into pCDH-PGK. pCDH-PGK-FLAG-Prdm16 or pCDH-PGK-tGFP, as control, were co-transfected together with second-generation packaging plasmids psPAX2 and pMD2.G into HEK 293T cells using PolyFect transfection reagent (Qiagen, # 301105) for production of lentiviral particles. Medium supernatant containing viral particles was collected at three consecutive days and concentrated using PEG-it Virus Precipitation Solution (Biocat, # LV810A-1-SBI). Physical virus titer was determined by One-Wash Lentivirus Titer Kit, HIV-1 p24 ELISA (Origene, #TR30038), and primary iWAT preadipocytes were transduced at 50% confluency with multiplicity of infection (MOI) 500 or 800 using polybrene (8 μg/ml) for 24 h. At 80% confluency, transduced preadipocytes were induced for 2 days and ASO-mediated Ctcflos KD was performed at the first day of differentiation as described above. Cells were harvested 72 h after transfection and analyzed for gene expression by qRT–PCR.

## Equivalence testing

Equivalence was tested by Two-One-Sided-Tests (TOST) Equivalence Testing for dependent *t*-test using the R function TOSTpaired.raw (significance level alpha = 0.1). Equivalence bounds (± 0.18) were set based on the multiplicative gene interaction model that postulates that the effect size of a double KD of two genes would be equal to the multiplication of their single KD effect sizes (0.36 × 0.51 = 0.18) (Fig 6H), if the two genes work completely independent of each other. Lower effect sizes point toward an epistasis of the two genes.

## Total and fractionated RNA isolation, poly-A RNA enrichment

For total RNA isolation, frozen tissues or cultured cells were homogenized and lysed in TRIsure (Bioline), RNA was purified on columns of SV total RNA isolation system (Promega) according to the manufacturer's protocol (including on column DNA digestion) and reverse transcribed into cDNA using SensiFast cDNA Synthesis Kit (Bioline). For fractionated isolation of nuclear, cytosolic, and

mitochondrial RNA, cells were lysed on ice in a high salt buffer (KCl (10 mM), MgCl$_2$ (1.5 mM), Tris (20 mM) pH7.5), homogenized by a dounce homogenizer (type B pestle, 0.0005–0.0025 in.) and after mixing with Triton X-100 (0.1%) centrifuged (1,000 *g*, 5 min) to pellet nuclei and other cell organelles as described previously (Rio *et al*, 2010). The cytosolic fraction in the supernatant was removed, and the pellet was resuspended in STE buffer (sucrose (250 mM), tris (5mM), EDTA (2 mM), pH 7.4) to further separate in a second centrifugation step (1,000 *g*, 3 min) mitochondrial (supernatant) and nuclear (pellet) fractions. RNA of the three fractions was then isolated as described for total RNA. Poly-A RNA enrichment was performed with Poly-ATtract mRNA isolation system (Promega) according to the manufacturer's instructions using biotinylated poly-dT primers that bind to transcript poly-A tails and can be captured by streptavidin-coupled paramagnetic particles.

## Gene expression analysis by quantitative real-time PCR (qRT–PCR) and PCR

Quantitative real-time PCR was performed with Lightcycler 480 II (Roche) in 384-well format (4titude). Genes of interest were amplified in triplicates using 1:2 diluted SensiMix SYBR No-ROX (Bioline), 250 μM gene-specific primers and 1 μl of 1:10 diluted cDNA template in a 12.5 μl reaction mix. Twofold serial dilutions of pooled cDNA samples served as standard curves for sample quantification. Transcript abundance was normalized to the expression of stable housekeeper gene *Gtf2b*. Forward (F) and reverse (R) primers used for qRT–PCR are listed below:

Gtf2b F: 5′-TGGAGATTTGTCCACCATGA-3′, R: 5′-GAATTGCCA AACTCATCAAAACT-3′;

Ctcflos tr1 F: 5′-ACTGTCCAGGTCCCTAACCC-3′, R: 5′-GACTCC CTGCTTGGAGAAGA-3′;

Ctcflos tr3+4b F: 5′-CCCCAGAACAAGAAGAGCTG-3′, R: 5′-CGA GTATTGGAGAAGGACAGC-3′;

Ucp1 F: 5′-GTACACCAAGGAAGGACCGA-3′, R: 5′-TTTATTCGT GGTCTCCCAGC-3′;

pre-Ucp1 F: 5′-GGATTGGCCTCTACGACTCA-3′, R: 5′-TAAGCTT CTCTGGGACCGTG-3′;

45SrRNA F: 5′-CTTGACCATGTCCCCAGAGT-3′, R: 5′-AGGCAC CTAGGAGACAAACC-3′;

12SrRNA F: 5′-ACACGACAGCTAAGACCCAA-3′, R: 5′-GGTAGA GCGGGGTTTATCGA-3′;

Cidea F: 5′-TGCTCTTCTGTATCGCCCAGT-3′, R: 5′-GCCGTGTT AAGGAATCTGCTG-3′;

Cox7a1 F: 5′-CCGACAATGACCTCCCAGTA-3′, R: 5′-TGTTTGTCC AAGTCCTCCAA-3′;

Atgl F: 5′-GGAAATTGGGTGACCATCTG-3′, R: 5′-AAGGCCACA TTGGTGCAG-3′;

Hsl F: 5′-GCTTGGTTCAAGTGGAGAGC-3′, R: 5′-GCCTAGTGCC TTCTGGTCTG-3′;

Fabp4 F: 5′-GATGGTGACAAGCTGGTGGT-3′, R: 5′-TTTATTT AATCAACATAACCATATCCA-3′;

Leptin F: 5′-TTCACACACGCAGTCGGTATC-3′ R: 5′-TGGTCCATC TTGGACAAACTCA-3′.

Prdm16 short F: 5′-GTACCCTGGCTTTGGACTCT-3′, R: 5′-GTAC CCTGGCTTTGGACTCT-3′ (forward primer spanning from exon 15 to 17), (reverse primer within exon 17) (Appendix Fig S7A);

Prdm16 long F: 5′-CTTTGGGAAGGGGCTGGAT-3′, R: 5′-GGTT AAAGGCTCCGGACTCT-3′ (forward primer within Prdm16 long specific exon 16), (reverse primer within exon 17) (Appendix Fig S7A);

Prdm16 total F: 5′-GTCAGAGGAGAAATTTGATGG-3′, R: 5′-AG AAGGAATGCTGTGAGTAG-3′; (forward and reverse primer within exon 9, present in Prdm16 short and long).

Ctcfl F: 5′-AGAGGACCCACAAGAACGAG-3′, R: 5′-AGTAGCT GCTTCTGTCGGAA-3′;

Pck1 F: 5′-GTGAGGAAGTTCGTGGAAGG-3′, R′-AGTGAGAGCC AGCCAACAGT-3′.

PCRs were conducted with 1:2 diluted ImmoMix (Bioline), including 500 nM gene-specific primers and 1 μl 1:10 diluted cDNA template in a total volume of 20 μl. PCRs were separated on a 2% agarose gel and visualized under UV light. Signal intensities were quantified with ImageJ software. Forward (F) and reverse (R) primers used for PCR are listed below:

Ctcflos tr1 F: 5′-GAGCTTGGAGGAAGTGAACG-3′, R: 5′-TCAAGG GACGATCTTTCCAG-3′;

Ctcflos tr2 F: 5′-GCAGTGAAACCTTCCAGAGC-3′, R: 5′-AAGT GATTCCAAGTGTCACCA-3′;

Ctcflos tr3+4a F: 5′-GCTGTCCTTCTCCAATACTCG-3′, R: 5′-GGG GTTAGGACAAAGAAGCTC-3′;

Ctcflos tr3+4b F: 5′-CCCCAGAACAAGAAGAGCTG-3′, R: 5′-CGAG TATTGGAGAAGGACAGC-3′;

Ctcflos tr4 F: 5′-CCCCAGAACAAGAAGAGCTG-3′, R: 5′-TTCT CTTCTCCCCACCCTTG-3′;

Prdm16 s + l F: 5′-GAGCCCCAAGGAGTCTATGA-3′, R: 5′-CGAG GGTCCTGTGATGTTCAA-3′ (exon 16 spanning primer pair to amplify both Prdm16 isoforms (Prdm16 long: 348bp; Prdm16 short: 173 bp) (Appendix Fig S6G));

Insr +/-exon 11 F: 5′-GAGGATTACCTGCACAACG-3′, R: 5′-TTCC TTTGGCTCTTGCCAC-3′.

## Lipid droplet staining and quantification

Lipid droplets of primary adipocytes were stained after fixation with formaldehyde (3.7%, 1 h) using Oil red O (0.3%, 1 h, Sigma-Aldrich). Cells were washed twice with PBS, and pictures were taken under the microscope with 10× and 32× objectives. For quantification, the cells were washed three times with isopropanol (60%, 5 min), Oil red O was extracted with 100% isopropanol (5 min), and absorbance was measured at 492 nm in a microplate luminometer (Infinite M200 Microplate reader, Tecan). Absorbance of blank 100% isopropanol was subtracted from absorbances of the samples.

Determination of lipid droplet numbers and size distributions was performed by automated digital image analysis (Wimasis).

## Lipolysis assay

Lipolysis rates were assessed by measurement of glycerol release under basal and isoproterenol-stimulated conditions. Primary cells were pretreated for 1 h and then incubated for another hour w/o isoproterenol (0.5 μM in DMEM with glucose (1g/l) and BSA (2%) pH 7.4, 1 h) and frozen for determination of protein concentration. The incubation medium was harvested, and 10 μl thereof was

incubated with free glycerol reagent (100 μl, Sigma-Aldrich) for 15 min in the dark to determine lipolysis-dependent release of glycerol into the medium. The quinoneimine dye produced in the color reaction is proportional to the glycerol concentration and was measured at 540 nm in a microplate luminometer (Infinite M200 Microplate reader, Tecan). Glycerol concentrations were normalized to protein amount.

## Mito tracker staining

Mitochondria were stained with Mito Tracker Deep Red 633 (100 nM in DMEM, 30 min, 37°C, Invitrogen), fixed with formaldehyde (3.7% in DMEM, 10 min, 37°C), mounted with Vectashield Mounting Medium with DAPI for nuclear counterstain (Vector Labs), sealed with a coverslip and nail polish, and visualized with 60× oil immersion objective in a confocal laser-scanning microscope (Olympus FluoView FV10i). Fluorescent signals were quantified with ImageJ software.

## Splicing inhibition

For splicing inhibition experiments, primary inguinal adipocytes of 129S6 mice were cultured according to the brite adipocyte differentiation protocol as described above. On the first day of differentiation, cells were treated with 10, 15, or 20 nM isoginkgetin (Tocris) diluted in DMSO or a corresponding amount of DMSO as control. After 48 h, cells were washed with PBS and harvested for qPCR analysis.

## Respirometry

Oxygen consumption of primary adipocytes was measured by an XF96 Extracellular Flux Analyzer (Seahorse Bioscience) as described in Li et al, (2017). Primary adipocytes were reverse transfected onto XF96 microplates at the first or fifth day of differentiation and subjected to respirometry 72 h later. Cells were incubated with Seahorse assay medium (DMEM basal medium with glucose (25 mM), sodium pyruvate (2 mM), NaCl (31 mM), GlutaMax (2mM) phenol red (15 mg/l) and essentially fatty acid-free bovine serum albumin (BSA) (2%) (pH 7.4) at 37°C in a room air incubator for 1 h. Basal oxygen consumption was determined of untreated cells, while consecutive injections of oligomycin (5 μM), isoproterenol (0.5 μM), FCCP (1 μM), and antimycin A (5 μM) allowed measurement of basal leak, UCP1-dependent uncoupled, maximal, and non-mitochondrial respiration, respectively. Oxygen consumption rates were calculated by the Seahorse XF96 software, and non-mitochondrial oxygen consumption was subtracted from all other values. UCP1-mediated uncoupled respiration was expressed as fold of isoproterenol respiration over oligomycin-dependent basal leak respiration.

## Western blot

For immunological determination of protein abundance, cells were lysed with radio immunoprecipitation assay (RIPA) buffer (NaCl (150 mM), Tris–HCl (50 mM), EDTA (1 mM) NP-40 (1%), and Na-deoxycholate (0.25%)), 30 μg of extracted protein was size separated in a 12.5% SDS–PAGE and blotted by a Trans-Blot SD semi-dry transfer cell (Bio-Rad) to a PVDF membrane (Millipore). Overnight blocking of the membrane with BSA was followed by treatment with

primary antibodies against UCP1 (rabbit anti-UCP1, Abcam AB23841, 1:10,000), COX4 (rabbit anti-COX4, Cell Signaling 4844 Technology, 1:5,000), and housekeeper β-Actin (mouse anti-Actinb, EMD Millipore MAB1501, 1:5,000). Fluorophore-conjugated secondary antibodies (goat anti-rabbit IRDye 800CW and donkey anti-mouse IRDye 680, Licor Biosciences, 1:20,000) were applied and fluorescent signals detected by Odyssey fluorescent imager (Licor Biosciences).

## Immunocytochemistry

For immunofluorescent staining of SC35/SRSF2 and UCP1 in primary adipocytes, cells were fixed with cooled ethanol (95%) and glacial acetic acid (5%) (10 min, −20°C), permeabilized by Triton X-100 (0.3%, 10 min), blocked with donkey serum (10% in PBST with Triton X-100 (0.3%), 2 h), incubated with primary antibodies against SC35 (mouse anti-SC35, Abcam ab11826, diluted 1:100 in PBST with Triton X-100 (0.3%) and donkey serum (1%), overnight) or against UCP1 (rabbit anti-UCP1, Abcam, diluted 1:200 in PBST with Triton X-100 (0.3%) and donkey serum (1%), overnight) and fluorophore coupled anti-mouse or anti-rabbit secondary antibodies (Alexa Fluor 647 donkey anti-mouse IgG, Life Technologies or Alexa Fluor 546 donkey anti-rabbit IgG, Life technologies, both diluted 1:500 in PBST with donkey serum (1%). Slides were mounted with Vectashield Mounting Medium with DAPI for nuclear counterstain (Vector Labs), sealed with a coverslip and nail polish, and visualized with 60× oil immersion objective in a confocal laser-scanning microscope (Olympus FluoView FV10i). Fluorescent signals were quantified with ImageJ software.

## Statistics

Values are presented as means ± SD or means and individual values. Significance of differences was assessed by paired or unpaired two-tailed Student's *t*-test for single comparisons, one-way or two-way ANOVA for comparison of two or more groups with Šídák post hoc test for multiple comparisons. Correlations are given with Pearson correlation coefficients. Statistical significance is considered for *P*-values below 0.05. Statistical analyses were conducted by GraphPad prism 6 software.

## Data availability

- RNA sequencing data of Ctcflos KD and control samples 24 or 72 h after knockdown are available at Gene Expression Omnibus GSE169150 (https://www.ncbi.nlm.nih.gov/geo/query/acc.cgi?acc = GSE169150).
- Deep RNA sequencing data of Ctcflos KD and control cells 24 h after knockdown are available at Gene Expression Omnibus GSE169151 (https://www.ncbi.nlm.nih.gov/geo/query/acc.cgi?acc = GSE169151).

**Expanded View** for this article is available online.

## Acknowledgements

We would like to thank Sebastian Dieckmann and Stefanie Maurer for providing iWAT cDNA samples of BL/6J mice housed at different temperatures, Alexandra Triebig for support with lentiviral overexpression, Katharina Schnabl, Josef Oeckl, and Thomas Gantert for scientific support and Sabine Mocek for excellent technical assistance. This research was supported by the German Research Society (KL 973/12-1, LI 3716/1-1), seed money from the German Obesity Competence Network (BMBF01GI1325), the Else Kröner-Fresenius-Stiftung (EKFS). Open Access funding enabled and organized by Projekt DEAL.

## Author contributions

AB-H, YL, and MK designed the study and wrote the manuscript. AB-H conducted experiments and analyzed data. CK helped with some experiments and performed data analysis. PW performed SGSeq analysis. AS helped with immunocytochemistry and Mito Tracker experiments. PCS and BD conducted *Ctcflos* KD RNA-seq experiments. TF provided iBAT RNA-seq data and helped with RNA-seq analysis.

## Conflict of interest

The authors declare that they have no conflict of interest.

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
