## [Review Process File · EMBO Reports]

LncRNA Ctcfls orchestrates transcription and alternative splicing in thermogenic adipogenesis

Andrea Bast-Habersbrunner, Christoph Kiefer, Peter Weber, Tobias Fromme, Anna Schießl, Petra Schwalie, Bart Deplancke, Yongguo Li, and Martin Klingenspor

DOI: [10.15252/embr.202051289](https://doi.org/10.15252/embr.202051289)

Corresponding author(s): Martin Klingenspor (mk@tum.de), Yongguo Li (yongguo.li@tum.de), Andrea Bast-Habersbrunner (andrea.bast@mytum.de)

Review Timeline:

Submission Date:	9th Jul 20
Editorial Decision:	25th Aug 20
Revision Received:	19th Mar 21
Editorial Decision:	20th Apr 21
Revision Received:	21st Apr 21
Accepted:	22nd Apr 21

Editor: Deniz Senyilmaz Tiebe

Transaction Report:

Dear Dr. Klingenspor,

Thank you for the submission of your research manuscript to our journal, which was now seen by three referees, whose reports are copied below.

My apologies for the delay in getting back to you, it took longer than anticipated to receive the referee reports.

We concur with the referees that proposed role of Ctcfls lncRNA in regulation of brite adipogenesis is in principle very interesting. However, referees also raise important concerns that need to be addressed to consider publication here. In particular,

1. Additional controls for ASO mediated Ctcfls depletion is required (ref #1 point 1, ref #2 standfirst and 1st major point).
2. More support for the proposed effect of Ctcfls on Prdm16 splicing and the claim that Ctcflc acts in cis is necessary (ref #1 points 2 and 3).
3. The epistasis between Ctcfls and Prdm16 needs to be strengthened (ref #1 point 4).

I find the reports informed and constructive, and believe that addressing the concerns raised will significantly strengthen the manuscript.

Given these positive recommendations, we would like to invite you to revise your manuscript with the understanding that the referee concerns (as in their reports) must be fully addressed and their suggestions taken on board. Please address all referee concerns in a complete point-by-point response. Acceptance of the manuscript will depend on a positive outcome of a second round of review. It is EMBO reports policy to allow a single round of revision only and acceptance or rejection of the manuscript will therefore depend on the completeness of your responses included in the next, final version of the manuscript.

*** Temporary update to EMBO Press scooping protection policy:

We are aware that many laboratories cannot function at full efficiency during the current COVID-19/SARS-CoV-2 pandemic and have therefore extended our 'scooping protection policy' to cover the period required for a full revision to address the experimental issues highlighted in the editorial decision letter. Please contact the scientific editor handling your manuscript to discuss a revision plan should you need additional time, and also if you see a paper with related content published elsewhere.***

1. A data availability section providing access to data deposited in public databases is missing (where applicable).
2. Your manuscript contains statistics and error bars based on $n=2$. Please use scatter plots in these cases.

Supplementary/additional data: The Expanded View format, which will be displayed in the main HTML of the paper in a collapsible format, has replaced the Supplementary information. You can submit up to 5 images as Expanded View. Please follow the nomenclature Figure EV1, Figure EV2 etc. The figure legend for these should be included in the main manuscript document file in a section called Expanded View Figure Legends after the main Figure Legends section. Additional Supplementary material should be supplied as a single pdf labeled Appendix. The Appendix includes a table of content on the first page with page numbers, all figures and their legends. Please follow the nomenclature Appendix Figure Sx throughout the text and also label the figures according to this nomenclature. For more details please refer to our guide to authors.

Please note that for all articles published beginning 1 July 2020, the EMBO Reports reference style will change to the Harvard style for all article types. Details and examples are provided at <https://www.embopress.org/page/journal/14693178/authorguide#referencesformat>

- 1) a .docx formatted version of the manuscript text (including legends for main figures, EV figures and tables). Please make sure that the changes are highlighted to be clearly visible.
- 2) individual production quality figure files as .eps, .tif, .jpg (one file per figure).
- 3) a .docx formatted letter INCLUDING the reviewers' reports and your detailed point-by-point responses to their comments. As part of the EMBO Press transparent editorial process, the point-by-point response is part of the Review Process File (RPF), which will be published alongside your paper. For more details on our Transparent Editorial Process, please visit our website: <https://www.embopress.org/page/journal/14693178/authorguide#transparentprocess>
You are able to opt out of this by letting the editorial office know (emboreports@embo.org). If you do opt out, the Review Process File link will point to the following statement: "No Review Process File is available with this article, as the authors have chosen not to make the review process public in this case."
- 4) a complete author checklist, which you can download from our author guidelines (<http://embor.embopress.org/authorguide>). Please insert information in the checklist that is also reflected in the manuscript. The completed author checklist will also be part of the RPF.
- 5) Please note that all corresponding authors are required to supply an ORCID ID for their name upon submission of a revised manuscript (<https://orcid.org/>). Please find instructions on how to link your ORCID ID to your account in our manuscript tracking system in our Author guidelines (<http://embor.embopress.org/authorguide>).
- 6) We replaced Supplementary Information with Expanded View (EV) Figures and Tables that are collapsible/expandable online. A maximum of 5 EV Figures can be typeset. EV Figures should be cited as 'Figure EV1, Figure EV2' etc... in the text and their respective legends should be included in the main text after the legends of regular figures.

- For the figures that you do NOT wish to display as Expanded View figures, they should be bundled together with their legends in a single PDF file called *Appendix*, which should start with a short Table of Content. Appendix figures should be referred to in the main text as: "Appendix Figure S1, Appendix Figure S2" etc. See detailed instructions regarding expanded view here: <http://embor.embopress.org/authorguide#expandedview>.

7) We would also encourage you to include the source data for figure panels that show essential data.

Numerical data should be provided as individual .xls or .csv files (including a tab describing the data). For blots or microscopy, uncropped images should be submitted (using a zip archive if multiple images need to be supplied for one panel). Additional information on source data and instruction on how to label the files are available <http://embor.embopress.org/authorguide#sourcedata>.

8) Our journal encourages inclusion of *data citations in the reference list* to directly cite datasets that were re-used and obtained from public databases. Data citations in the article text are distinct from normal bibliographical citations and should directly link to the database records from which the data can be accessed. In the main text, data citations are formatted as follows: "Data ref: Smith et al, 2001" or "Data ref: NCBI Sequence Read Archive PRJNA342805, 2017". In the Reference list, data citations must be labeled with "[DATASET]". A data reference must provide the database name, accession number/identifiers and a resolvable link to the landing page from which the data can be accessed at the end of the reference. Further instructions are available at <http://embor.embopress.org/authorguide#datacitation>.

9) Please make sure to include a Data Availability Section before submitting your revision - if it is not applicable, make a statement that no data were deposited in a public database. Primary datasets (and computer code, where appropriate) produced in this study need to be deposited in an appropriate public database (see <http://embor.embopress.org/authorguide#dataavailability>).

The accession numbers and database should be listed in a formal "Data Availability " section (placed after Materials & Method) that follows the model below. Please note that the Data Availability Section is restricted to new primary data that are part of this study.

Data availability

10) Regarding data quantification, please ensure to specify the name of the statistical test used to generate error bars and P values, the number (n) of independent experiments underlying each data point (not replicate measures of one sample), and the test used to calculate p-values in each figure legend. Discussion of statistical methodology can be reported in the materials and methods section, but figure legends should contain a basic description of n, P and the test applied.

Please note that error bars and statistical comparisons may only be applied to data obtained from at least three independent biological replicates.

I look forward to seeing a revised version of your manuscript when it is ready. Please let me know if you have questions or comments regarding the revision.

Yours sincerely,

Deniz Senyilmaz Tiebe

Deniz Senyilmaz Tiebe, PhD
Editor
EMBO Reports

Referee #1:

In the manuscript the authors study the functions of Ctcfls, a previously unstudied lncRNAs in the biology of adipogenesis. The authors show a phenotype in cells where Ctcfls has been perturbed, then show the molecular signature of loss of Ctcfls is related to that of loss of Prdm16, and that some changes in alternative splicing are also potentially involved, possibly indirectly. Overall, the discovery of a lncRNA functioning in adipose cell biology is of interest to researchers interested in brite adipogenesis. Beyond the molecular consequences of differentiation, the insight into what Ctcfls actually does is quite limited, and some of the conclusions do not appear to be supported by significant changes in the data or are somewhat over-interpreted. Specifically, whether Ctcfls affects splicing Prdm16 is not clear. Overall the conclusion that "our findings emphasize the mechanistic versatility of lncRNAs acting at several independent levels of gene expression for effective regulation of key differentiation factors to direct cell fate and function." is not really supported, as it is not clear what Ctcfls affects directly (it could just regulate one gene), and certainly not clear if it is indeed affecting multiple different layers. The work could benefit by additional experiments supporting the findings and more careful data interpretation.

Major comments

1. The presented results on Ctcfls loss-of-function are based on the use of ASOs, which are a

valid tool for perturbing lncRNAs, but also suffer from potential off-target or toxicity effects. Therefore it is usually required to show similar results using another technique, such as CRISPR editing, CRISPR interference or even just another ASO targeting the same isoform. This is important for showing that the observed molecular phenotypes are indeed robust. Also, if the authors think that the lncRNA is not acting in cis, then they can try to rescue the phenotype by over-expressing the lncRNA. If such a rescue experiment is technically particularly challenging that should be mentioned/discussed.

2. It is not clear how strong is the effect of loss of Ctcfl on Prdm16 splicing. When changes in expression levels are involved, it is difficult to obtain convincing changes in splicing. The authors rely on some quantification from an agarose gel (Fig. 6K), which is noisy and not really quantitative, and also prone to the challenges of different expression levels. In the RNA-seq data, what does splicing of Prdm16 look like? Is there a significant change observed there? The authors must show what the read coverage in this region looks like in their RNA-seq data to show whether Ctcfl inhibition indeed substantially affects splicing of Prdm16.

3. The evidence that Ctcfl does not act in cis is not really shown. Can the authors show RNA-seq read coverage for the whole locus to show that Ctcfl is indeed not expressed in their cells? For Pck1, which is a suspect target because it is a gene involved in metabolism, the data for its down-regulation in Ctcfl should be explicitly shown (e.g., from the RNA-seq data the authors have), as should be shown the data that Pck1 knockdown does not affect Ucp1 expression, which is now only mentioned in the paper (lines 320-321).

4. The claim of an epistasis between Ctcfl and Prdm16 based on figure 5H is quite weak. The differences seem to be small, and it appears that the experiment is not really powerful enough to show that the combination of Ctcfl ASO1 and siPrdm16 gives an effect similar to just one of them. Can the authors rescue the phenotype of Ctcfl ASO1 by over-expressing Prdm16? That would be far more convincing. Otherwise, they should provide evidence that there is a significant similarity between siPrdm16+Ctcfl ASO1 and the individual KDs of Ctcfl and Prdm16. Also, why is knockdown of Prdm16 so inefficient? (Fig. 5J)

5. Throughout the paper there are various changes shown, and for some of them it's not clear if they are significant or not, although they seem to be treated as such. For example Fig. 4L, 4S

Minor comments:

1. Line 173: The finding that lncRNAs are differentially expressed does not support the "initial hypothesis about their regulatory involvement". Genes can be differentially expressed yet be completely uninvolved in regulation / non-functional. This statement should be toned down. Same for line 193: "substantiated the involvement of positively regulated lncRNAs in the browning process".

2. Is there any evidence that Ctcfl is conserved in human? This should be mentioned/discussed.

Referee #2:

The authors propose the lncRNA Ctcfl as a regulator of adipose browning, and perform an impressive number of experiments and analysis to characterize its function within this context. Their results suggest that the main role of Ctcfl in browning is through regulation of expression and alternative splicing of Prdm16, a well known regulator of brown adipocyte development.

This work is a part of the important but gigantic task of characterizing the function of non-coding RNA in the mammalian genome. Although several efforts have been made to establish function for non-coding RNA at a genome wide level, it is becoming increasingly clear that these efforts need to be complemented by deep studies of individual loci in specific cellular contexts. The authors present

here one example of this laborious but much needed work.

The manuscript also illustrates the difficulties involved in determining non-coding RNA function. These transcripts are usually lowly expressed, unstable, and may exist in many unannotated forms. Designing appropriate knock-down or over-expression experiments is not an easy task and there are many pitfalls.

The manuscript is clear and easy to follow. Most conclusions are supported by data but there is a design flaw in the ASO experiments: relying on only one ASO per isoform when knocking down Ctcfls. I detail my concerns below.

Major:

* A design flaw in the study is that only one ASO was used for knock-down of each isoform of Ctcfls. When using oligo based knock-down (ASOs, shRNA, siRNA etc) it is widely recommended to use multiple independent oligos targeting the molecule of interest, in order to minimize the risk that the observed effects of the knock-down wholly or partly result from off-target effects rather than the actual knock-down. In the current study, a better design would have been to use at least one more ASO per isoform in parallel, in order to exclude this possibility. The problem is somewhat mitigated in some experiments where two different isoforms ("tr1" and "tr3") are knocked down by different ASOs ("ASO1" and "ASO2" respectively), and it is promising that the two different ASOs seem to yield (mostly) similar results. However, for many experiments the authors abandon this approach and rely solely on results using ASO1. That leaves the possibility that some of the following observations are due to off-target effects. Best would be to redo these experiments with a different ASO, but at the very least, text should be added to the manuscript that clearly discusses the possibility that some of the ASO knockdown results may be due to off-target effects.

* The authors conclude from their experiments that Ctcfls has a dual function: modulating the transcription levels as well as isoform ratios of Prdm16. But whereas possible mechanisms through which Ctcfls may influence the splicing machinery are discussed at length in the Discussion section, no similar suggestions for how Ctcfls may regulate transcript levels are introduced in the manuscript. In fact, the experiments are entirely consistent with a less complicated explanation: that altered transcription levels follows from altered splicing. Unless evidence is presented that there are two distinct mechanisms (or at least extensive discussion detailing this reasoning), it seems unwarranted to put so much emphasis on this dual function hypothesis.

Minor:

* In Fig 1B, I don't understand the meaning of the legend: what is "Contrib" and "Condition"?

* Fig 4J,K,L, the results regarding lipid droplet size: are any differences between NC and ASO1 KD significant? If not, these figures and corresponding text could be dropped or moved to supplement.

* Fig 5E, what was used as the background gene set in GO analysis? All genes, or all genes expressed in adipose tissue, or something else?

* Fig 5F, it is not easy to understand how to interpret the "Path Visio" figure and what it actually tries to visualize.

* Fig 6A, why did these terms not show up among terms discovered in Fig 5E? Or Fig 5E only

contains a selected set of GO terms? Please clarify.

* As far as I can see, Fig. 6 I is not referred to in the text.

Referee #3:

Bast-Habersbrunner et al. introduce the lncRNA Ctcflos as novel regulator of brite adipogenesis in vitro. Using big data analysis of lncRNA-expression profiles of adipose tissues from several inbred mouse strains with varying degrees of adipose browning Ctcflos emerges as a candidate of interest. Ctcflos is shown to affect adipocyte differentiation on the levels of transcription and alternative splicing, exemplified by Ucp1 and Prdm16 mRNAs. Furthermore, the authors demonstrate altered adipocyte function upon knockdown of Ctcflos.

The research is novel, technically sound and the manuscript is very well prepared. To my opinion, this manuscript has no major weaknesses.

Minor Points:

1. This work originates from the study of inguinal white adipose tissue, which is an adipose depot that readily undergoes adipose browning. Hence, it is logical that the authors chose to study the role of Ctcflos in what they refer to as brite adipocytes. However, in the end, the data also allow drawing the conclusion that also in classical brown adipocytes, e.g. derived from the interscapular depot, Ctcflos is expressed, regulated and has functional impact. To my opinion, this could be highlighted better.
2. The references to Figure S4P&Q in lines 316-323 is missing.
3. Figure 6B shows a decrease of Ucp1 mRNA per Fabp4 upon isoginkgetin: it would be nice to also show the levels of both genes in comparison to a housekeeping gene.
4. In line 807 it appears that blocking was done with TritonX-100 and not "TiritonX-100"
5. The legend in Figure 1B (not the figure legend) does not explain well what is actually shown in that PCA plot.
6. In Figure 4F please specify the loading control of the Western Blot shown in the figure legend.
7. The panels in Figure 6C should be enlarged for better visibility.

Answers to the Referee comments

Referee #1:

In the manuscript the authors study the functions of *Ctcflos*, a previously unstudied lncRNAs in the biology of adipogenesis. The authors show a phenotype in cells where *Ctcflos* has been perturbed, then show the molecular signature of loss of *Ctcflos* is related to that of loss of *Prdm16*, and that some changes in alternative splicing are also potentially involved, possibly indirectly. Overall, the discovery of a lncRNA functioning in adipose cell biology is of interest to researchers interested in brite adipogenesis. Beyond the molecular consequences of differentiation, the insight into what *Ctcflos* actually does is quite limited, and some of the conclusions do not appear to be supported by significant changes in the data or are somewhat over-interpreted. Specifically, whether *Ctcflos* affects splicing *Prdm16* is not clear. Overall the conclusion that "our findings emphasize the mechanistic versatility of lncRNAs acting at several independent levels of gene expression for effective regulation of key differentiation factors to direct cell fate and function." is not really supported, as it is not clear what *Ctcflos* affects directly (it could just regulate one gene), and certainly not clear if it is indeed affecting multiple different layers. The work could benefit by additional experiments supporting the findings and more careful data interpretation.

Major comments

1. The presented results on *Ctcflos* loss-of-function are based on the use of ASOs, which are a valid tool for perturbing lncRNAs, but also suffer from potential off-target or toxicity effects. Therefore it is usually required to show similar results using another technique, such as CRISPR editing, CRISPR interference or even just another ASO targeting the same isoform. This is important for showing that the observed molecular phenotypes are indeed robust. Also, if the authors think that the lncRNA is not acting in cis, then they can try to rescue the phenotype by over-expressing the lncRNA. If such a rescue experiment is technically particularly challenging that should be mentioned/discussed.

We thank the reviewer for this comment and completely agree that it is of prime importance to convincingly show that the described effects of *Ctcflos* knockdown (KD) on *Ucp1* gene expression and the resulting phenotype do not occur as a consequence of an off-target effect of the applied antisense oligonucleotides (ASOs). As suggested by the reviewer we therefore repeated the knockdown of *Ctcflos* with another four ASOs, two ASOs targeting *Ctcflos* isoform 1 (ASO 3, 4) and two ASOs targeting isoform 3 and 4 (ASO 5, 6). Comparable to the previously presented effects of ASOs 1 and 2, knockdown of *Ctcflos* by the four new ASOs also led to an impairment of *Ucp1* gene transcription (Figure below) (revised manuscript Fig. 4 C-E).

Figure legend-Knockdown of *Ctcflos* transcript 1, 3 and 4 by further LNA Gapmer ASOs. (A, B) Efficiency of

Ctcflos KD (A) by new ASOs 3 and 4 alongside previously shown ASO1 targeting *Ctcflos* transcript 1 and (B) by new ASOs 5 and 6 alongside previously shown ASO2 targeting *Ctcflos* transcript 3 and 4. (C) Impact of *Ctcflos* KD on *Ucp1* gene transcription including new and previously characterized ASOs.

As proposed by the reviewer, a rescue experiment by *Ctcflos* overexpression would indeed be a valuable strategy to demonstrate that *Ctcflos* acts in trans. Overexpression of *Ctcflos* is however challenging as our RNA sequencing data do not entirely match with the published annotations of *Ctcflos* transcripts. Reads mapping outside annotated exon boundaries hint towards prolonged transcript 3' ends as likewise predicted by de novo reconstruction of the *Ctcflos* transcripts based on our RNA sequencing data (Figures below).

Figure legend-Read mapping to the *Ctcflos* locus in differentiated brite adipocytes. Reads mapped to the *Ctcflos* locus in RNA sequencing of three biological replicates of brite adipocytes at day 3 of differentiation. Red boxes highlight read mapping outside of *Ctcflos* exon boundaries proposing prolonged 3' transcript ends. Genomatix Software Suite.

Annotation

De novo reconstruction

Figure legend-Annotated and de novo reconstructed *Ctcflos* transcripts. Comparison of Ensembl annotated *Ctcflos* isoforms and *Ctcflos* transcripts predicted from RNAseq read mapping using StringTie de novo reconstruction. Red boxes highlight prolonged 3' ends of *Ctcflos* exons.

Also for *Ctcflos* isoform 1 (ENSMUST00000144256.1; AK041254), which would be the favored isoform for overexpression, de novo reconstruction questions the Ensembl annotation. Although we could confirm by PCR that exon 1 and 2 are correctly annotated, our data predict a prolonged exon 3. 3' rapid amplification of cDNA ends (3'RACE) could not yet provide reliable information about the 3' end of the transcript. Due to these uncertainties, plasmid-based gene transfer might not achieve to deliver the correct and functional *Ctcflos* transcript that is present in brite adipocytes. In order to overexpress the brite cell characteristic *Ctcflos* transcripts, will therefore prefer to use a CRISPR activation (CRISPRaSAM) system in future that aims to enhance the expression of the endogenous target gene by guide RNA-mediated recruitment of deactivated, transcription activator coupled Cas9 protein to the target gene promoter (Lundh *et al*, 2017).

A rescue experiment would aim to reverse the observed ASO-mediated *Ctcflos* knockdown phenotype by parallel or proximate overexpression of *Ctcflos*. To avoid that the *Ctcflos* transcripts produced by overexpression are also affected by the ASOs, this would however require the overexpression of a mutated version of *Ctcflos* that cannot be targeted by the ASOs but remains its functionality. Since we are both, unsure about the annotation of *Ctcflos* and do not know which part of the *Ctcflos* transcript conveys its functionality, this is particularly challenging for the current MS. These thoughts are included in the revised manuscripts discussion lines 527-531.

2. It is not clear how strong is the effect of loss of *Ctcflos* on *Prdm16* splicing. When changes in expression levels are involved, it is difficult to obtain convincing changes in splicing. The authors rely on some quantification from an agarose gel (Fig. 6K), which is noisy and not really quantitative, and also prone to the challenges of different expression levels. In the RNA-seq data, what does splicing of *Prdm16* look like? Is there a significant change observed there? The authors must show what the read coverage in this region looks like in their RNA-seq data to show whether *Ctcflos* inhibition indeed substantially affects splicing of *Prdm16*.

As depicted in the reviewer's comment, we observe two overlaying effects on *Prdm16* gene expression, (1) reduced *Prdm16* gene transcription and (2) altered *Prdm16* pre-mRNA splicing. The change in the overall expression level of *Prdm16* renders it indeed more difficult to deduce the effect on *Prdm16* alternative splicing. The *Ctcflos* KD effect on *Prdm16* transcription is quantified in the RNAseq data and by qPCR (Fig. 5 F and Fig. S5 G). Total *Prdm16* transcript levels are strongly reduced directly (24 hours) after the knockdown and remain to be diminished after 72 hours.

To analyze the *Ctcflos* KD effect on alternative splicing of *Prdm16*, we applied two approaches, (1) a more qualitative analysis by PCR that allows a direct comparison of the relative *Prdm16* short and long isoform abundances through visualization and (2) a quantitative analysis by qPCR that provides information on the effect strength.

We agree that quantification from an agarose gel picture is problematic and should not be used to estimate the effect size of splicing changes. Nevertheless, PCR analysis provides the advantage that both isoforms can be amplified and distinguished using the same primer pair. Both, short and long isoforms are thus amplified with the same efficiency, allowing a direct comparison with each other. In this respect, the agarose gel visualizes that signal strengths of *Prdm16* short and long strongly differ (clearly stronger signal for *Prdm16* short) in the control cells, while this difference is weakened in the *Ctcflos* KD cells (**Figure below B**) (Fig. 6 K, M). Despite the overlaying effect on *Prdm16* transcription, this change in the signal strengths of the two isoforms relative to each other demonstrates a shift in alternative splicing away from dominant *Prdm16* short to a more balanced splicing of both isoforms. In case of a mere transcriptional impact on *Prdm16* without alternative splicing changes, both signals (short and long) would be equally reduced by the KD. The PCR analysis however reveals, that only the *Prdm16* short signals but not the *Prdm16* long signals decrease.

To exclude the imprecisions of the agarose gel and to quantify the KD impact on *Prdm16* alternative splicing, we repeated the analysis by qPCR using specific primers for each isoform (**Figure below A**) (revised manuscript Fig. 6 J), 24 hours and 72 hours after *Ctcflos* KD (**Figure below C-E and H-J**) (revised manuscript Fig. 6 O-Q and T-V). Analogous to the observation in PCR analysis, despite overall reduction of *Prdm16* expression, the impact on *Prdm16* short is stronger than on *Prdm16* long: 24 hours after *Ctcflos* KD by ASO 1 there is a mean fold change (NC/ASO 1) of 4.90 of *Prdm16* short, which is significantly higher than the mean fold change (NC/ASO1) of 1.87 of *Prdm16* long (**Figure below F**) (revised manuscript Fig. 6 R). Similarly, 72 hours after *Ctcflos* KD by ASO1 the mean fold change (NC/ASO1) for *Prdm16* short is significantly greater than for *Prdm16* long, 1.83 vs. 1.08, respectively (**Figure below K**) (revised manuscript Fig. 6 W).

To further strengthen these results we repeated the experiment using ASO 2 (**Figure below C-E, G, H-J, L**) (revised manuscript Fig. 6 O-Q, S, T-V, X). Although there is just a non-significant trend 24 hours after the KD by ASO 2 for a greater impact on *Prdm16* short, we likewise observe a stronger reduction of *Prdm16* short compared to long 72 hours after *Ctcflos* KD (mean fold change (NC/ASO2) 2.55 and 1.44 for *Prdm16* short and long, respectively). These data thus further support that *Ctcflos* KD by ASO1 and ASO2 provokes a shift in the *Prdm16* alternative splicing profile, the strength of which can be extracted from the qPCR data.

Figure legend-Impact of *Ctflos* KD on *Prdm16* alternative splicing evaluated by PCR and qPCR. (A) Splice graph of *Prdm16* and resulting *Prdm16* long and short isoforms. Primers used to amplify both *Prdm16* isoforms at once by PCR are displayed in orange. Primers used to amplify specifically *Prdm16* short and long isoforms in qPCR are displayed in blue. (B) Gel picture of PCR products using exon 16 spanning primers to detect *Prdm16* long (upper band (348 bp)) and *Prdm16* short (lower band (173 bp)) comparing *Ctflos* tr1 (ASO 1) KD and control samples. (C-L) Effect of *Ctflos* KD on *Prdm16* isoform expression evaluated by qPCR. (C-E) Relative *Prdm16* total, short and long transcript levels after 24 hours in response to *Ctflos* KD by ASO1 and ASO2 compared to nontargeting controls. (F, G) Comparison of fold changes after 24 hours between control and *Ctflos* KD cells ((F) KD by ASO1, (G) KD by ASO 2) for *Prdm16* short and long isoform, respectively. (H-J) Relative *Prdm16* total, short and long transcript levels after 72 hours in response to *Ctflos* KD by ASO1 and ASO2 compared to nontargeting controls. (K, L) Comparison of fold changes after 72 hours between control and *Ctflos* KD cells ((K) KD by ASO1, (L) KD by ASO 2) for *Prdm16* short and long isoform, respectively. Mean values \pm SD, n=3-4, unpaired t tests, n.s. $p > 0.05$, * $p < 0.05$, ** $p < 0.01$ *** $p < 0.001$.

The search for confirmation of *Ctflos*-dependent *Prdm16* alternative splicing in the RNA seq raw data (24 hours after *Ctflos* KD), as proposed by the reviewer, is a reasonable suggestion. As the mapped reads originate from both isoforms and there is again the overlaying impact on total *Prdm16* expression it is, however, difficult to directly draw a conclusion on the KD effect on *Prdm16* alternative splicing. Regarding the entire *Prdm16* locus the reduction of total *Prdm16* transcript levels is visible (Figure below A upper panel). Zooming into the area of the alternatively spliced exon 16, which is present in the long and absent in the short isoform, the same can be observed (overall less reads map to this region in *Ctflos* KD samples) again due to overall reduction of *Prdm16* transcription (Figure below A lower panel). In order to draw conclusions on the abundances of short and long isoforms, respectively, intron-spanning reads that protrude beyond exon boundaries need to be extracted from the data. Reads spanning from exon 15 into exon 17 originate from *Prdm16* short, while reads spanning from exon 15 into exon 16 or from 16 into exon 17 support the long isoform. In Figure B below the depicted numbers count how many reads support the respective splice event (in control versus KD samples 4, 1 and 5 reads versus 2, 3 and 4 reads support the excision of exon 16 (short *Prdm16*), while 0, 0, 0 versus 0, 2, 1 reads support the splicing of exon 16 to exon 17 (long *Prdm16*)). Overall the number of reads supporting either *Prdm16* short or long is very limited and does not allow a quantitative evaluation. The sequencing depth of the RNAseq experiment was unfortunately too low to draw information on *Prdm16* splicing. Deeper sequencing of the *Prdm16* locus to investigate the phenomenon in future would indeed be valuable. The example of Insulin receptor, however, which is also *Ctflos* dependently spliced and for which RNAseq and PCR analysis provide the same results, support that the experimental evaluation of *Ctflos*-dependent alternative splicing is reliable. Additionally, the effect on *Prdm16* splicing was observed by two independent experimental settings (PCR and qPCR) using different primer pairs, further supporting the reliability of the results.

A

B

Figure legend-Information on *Ctcflos* KD dependent isoform profiles of *Prdm16* within deep RNA sequencing data. (A) Upper panel: Read mapping to the entire *Prdm16* locus in control (upper three lines) and *Ctcflos* KD cells (lower 3 lines). Lower panel: Read mapping to *Prdm16* exon 15, 16 and 17 in control (upper three lines) and *Ctcflos* KD cells (lower 3 lines). Genomatix Software Suite. (B) Splice graph of *Prdm16* generated based on intron-spanning reads and GencodevM21 exon annotation using SGSeq algorithm (Goldstein *et al*, 2016). Read mapping

of control (upper three lines) and *Ctcflos* KD (ASO1) cells (lower three lines). Numbers above or below the bended lines specify the amount of intron-spanning reads that support the depicted splice event. The red box highlights splice events involving exon 15, 16 and 17 and thus discriminating *Prdm16* short and long isoforms.

3. The evidence that *Ctcflos* does not act in cis is not really shown. Can the authors show RNA-seq read coverage for the whole locus to show that *Ctcflos* is indeed not expressed in their cells? For *Pck1*, which is a suspect target because it is a gene involved in metabolism, the data for its down-regulation in *Ctcflos* should be explicitly shown (e.g., from the RNA-seq data the authors have), as should be shown the data that *Pck1* knockdown does not affect *Ucp1* expression, which is now only mentioned in the paper (lines 320-321).

Below we show RNAseq read mapping to the locus of *Ctcflos* including its neighbor genes *Ctcf1* and *Pck1* in preadipocytes (lines 1-3) and differentiated brite adipocytes (lines 4-6) (Figure below A). *Ctcflos* and *Pck1* are, as expected, upregulated in the course of differentiation (note automatically adapted scales for each track). Compared to *Ctcflos* and *Pck1*, *Ctcf1* seems to be negligibly low expressed as no exon hitting reads are detected within its locus in the given scale ranges. Nevertheless, zooming into the *Ctcf1* specific region revealed that also *Ctcf1* expression is slightly upregulated during brite adipogenesis, with a low number of reads mapping to its locus in the differentiated state (Figure below B). Therefore, we tested by PCR whether we can detect *Ctcf1* experimentally at three different time points (in the preadipocyte state, at day1 and day 7 of differentiation) using two different primer pairs (spanning exon 8 and 9, or exon 10 and 11, respectively). Applying a high number of cycles (38) we observed a light upregulation of *Ctcf1* expression in the course of differentiation in accordance with the transcriptome analysis (Figure below C). At day 1 of differentiation, the time point at which we perform the *Ctcflos* KD, however, *Ctcf1* is very low expressed, though detectable. Also in deep RNA sequencing 24 hours after *Ctcflos* KD, *Ctcf1* was detected at low level (26-55 reads) (Figure below D, upper three lines). These data additionally predict that *Ctcflos* KD affects *Ctcf1* expression, being almost completely blunted in the absence of *Ctcflos* (Figure below D, lower three lines and E). This downregulation of *Ctcf1* might be an artefact of very low transcript abundance and rarely of biological relevance, nevertheless, we experimentally checked the impact of *Ctcflos* KD on *Ctcf1* expression and performed *Ctcf1* knockdown by a Dicer Substrate short interfering RNA (DsiRNA) to make sure that *Ctcf1* does not play any role in *Ucp1* gene expression. QPCR analysis of *Ctcflos* KD revealed no significant impact on *Ctcf1* gene expression levels (Figure below F) (revised manuscript Fig EV2Q) and knockdown of *Ctcf1* did not significantly affect *Ucp1* gene expression (Figure below G, H) (revised manuscript Fig EV2R and S), excluding *Ctcf1* as downstream mediator of the *Ctcflos* effects on *Ucp1* gene expression. These results are included in the revised manuscript in lines 328-336.

A

Figure legend-Regulation of *Ctcfl* gene transcription and its implication in the regulation of *Ucp1* gene expression. (A) Read mapping to the *Ctcfl* locus and its neighbor genes *Ctcfl* and *Pck1* in preadipocytes (upper three lines) and differentiated brite cells (lower three lines). (B) Read mapping to exons 6-11 of *Ctcfl* in preadipocytes (upper three lines) and differentiated brite cells (lower three lines). (C) Gel picture of PCR products using *Ctcfl* specific primers amplified from proliferating and differentiating brite adipocytes (day 1 and day 7 of differentiation). (D) Read mapping to exons 6-11 of *Ctcfl* in control (upper tree lines) and *Ctcfl* KD (ASO1) cells (lower three lines). (E, F) *Ctcfl* expression in control and *Ctcfl* KD (ASO1) cells (E) in deep RNA sequencing data and (F) assessed by qPCR. Mean values \pm SD, n=3, unpaired t tests, n.s. p>0.05, *p<0.05. (G) Efficiency of *Ctcfl* KD by dsiRNA. (H) Impact of *Ctcfl* KD on *Ucp1* expression assessed by qPCR. Mean values \pm SD, n=3, unpaired t tests, n.s. p>0.05, *p<0.05. Genomatrix Software Suite.

Gene expression of *Pck1*, the downstream neighbor of *Ctcfl*, is downregulated in *Ctcfl* deficiency (Figure below A, B), knockdown of *Pck1*, however, does not affect *Ucp1* transcription (Figure below C, D), excluding also *Pck1* as mediator of the *Ctcfl* KD effect in brite adipogenesis. The data supporting this are shown in the supplement data (previous Fig. S4 P-R) (revised manuscript Fig. T-V).

Figure legend - Regulation of *Pck1* gene transcription and its implication in the regulation of *Ucp1* gene expression. (A) Read mapping to the *Pck1* locus in control (upper tree lines) and *Ctcflos* KD (ASO1) cells (lower three lines). Genomatix Software Suite. (B) *Pck1* expression in control and *Ctcflos* KD (ASO1) cells assessed by qPCR. Mean values \pm SD, n=3, unpaired t test, ***p<0.001. (C) Efficiency of *Pck1* KD by siRNA. (H) Impact of *Pck1* KD on *Ucp1* expression assessed by qPCR. Mean values \pm SD, n=1 (6 technical replicates), unpaired t tests, n.s. p>0.05, ***p<0.0001.

4. The claim of an epistasis between *Ctcflos* and *Prdm16* based on figure 5H is quite weak. The differences seem to be small, and it appears that the experiment is not really powerful enough to show that the combination of *Ctcflos* ASO1 and *siPrdm16* gives an effect similar to just one of them. Can the authors rescue the phenotype of *Ctcflos* ASO1 by over-expressing *Prdm16*? That would be far more convincing. Otherwise, they should provide evidence that there is a significant similarity between *siPrdm16*+*Ctcflos* ASO1 and the individual KDs of *Ctcflos* and *Prdm16*. Also, why is knockdown of *Prdm16* so inefficient? (Fig. 5J)

We thank the reviewer for the suggestion to perform a rescue experiment by *Prdm16* overexpression in *Ctcflos* KD cells to strengthen our hypothesis that *Prdm16* plays a role in the mediation of *Ctcflos* function in brite adipogenesis. We followed this advice, conducted *Ctcflos* KD in *Prdm16* overexpressing cells, and compared the resulting phenotype to that of non-*Prdm16* overexpressing control and *Ctcflos* KD cells. We observed that *Prdm16* overexpression in *Ctcflos* deficiency could rescue *Ucp1* gene expression, reaching or exceeding *Ucp1* transcript levels of control cells (see Figure below). *Prdm16* was overexpressed by lentiviral infection in proliferating primary iWAT cells (using multiplicity of infection (MOI) ratios of 500 to 800). GFP lentivirus overexpressing (GFP) cells (A-C) served as controls. To avoid that variations in differentiation between the treatment groups distort the analysis, we normalized the gene expression data by adipocyte differentiation marker fatty acid binding protein 4 (FABP4). In an alternative experiment *Prdm16* overexpressing cells were compared with uninfected (untr) controls, providing similar results (D-F). Overall, the observation that *Prdm16* overexpression abrogates the impact of *Ctcflos* KD on *Ucp1* gene expression in independent experiments supports the epistasis between *Ctcflos* and *Prdm16*. Nevertheless, we do not want to claim that the function of *Ctcflos* in brite adipogenesis relies entirely on *Prdm16* but is most likely more complex. With its impact on *Prdm16* transcription and alternative splicing, *Ctcflos* however modulates a key factor in brite adipogenesis, which, as supported by the presented data, most likely contributes to a large extend to the observed *Ctcflos* KD phenotype.

Figure legend (data now included in Fig. 5 K-M and Fig. EV3 F-H) – *Prdm16* overexpression rescues the effect of *Ctcflos* KD on *Ucp1* gene transcription. Primary iWAT cells were either untreated, infected with *Prdm16*- or turbo green fluorescent protein (GFP)-expressing viral particles at the second day of proliferation. This was followed by reverse transfection at the first day of differentiation using nontargeting control (NC) or *Ctcflos*-targeting ASO1. Gene expression was analyzed 72 hours later by qPCR. Mean and individual values. Each graph presents pooled data from two experiments slightly varying virus titers. RM One-way ANOVA (Tukey-test), n.s. $p > 0.05$, * $p < 0.05$, ** $p < 0.01$, *** $p < 0.001$, **** $p < 0.0001$.

Figure legend – Microscopic images of differentiating brite adipocytes after lentiviral infection and *Ctcflos* KD or respective control treatments.

Additionally, as proposed in the reviewer comment we analyzed the combined *Prdm16*-*Ctcflos* KD experiment for equivalence between single and double knockdown of *Ctcflos* and *Prdm16*.

For this purpose we performed a TOST test, assuming that two groups are equivalent if their difference is small enough to not exceed upper and lower bounds of equivalence that we set based on the multiplicative gene interaction model according to (Costanzo *et al*, 2019). The effect size that would be expected for the double knockdown, if *Ctcflos* and *Prdm16* would work completely independent of each other, thereby results from the multiplication of the single knockdown effects (effect size of single KD *Ctcflos* on *Ucp1* gene expression=**0.36**; effect size of single KD *Prdm16* on *Ucp1* gene expression=**0.51**; theoretically expected effect size of double KD, assuming independent activities for *Ctcflos* and *Prdm16* = $0.36 \times 0.51 = 0.18$; measured effect size of double knockdown = 0.26). If the difference between single and double knockdown does not exceed $0.36 - 0.18 = 0.18$, this would argue for a dependency/epistasis between *Ctcflos* and *Prdm16*. Based on this, the upper and lower bounds of equivalence in the TOST test were set to 0.18 and -0.18, respectively. At a significance level of $\alpha=0.1$ (but not $\alpha=0.05$) it can be rejected that the difference between single (*Ctcflos*) and double (*Ctcflos+Prdm16*) knockdown is large enough to be considered, or in other words, that there is a significant similarity between the two groups (Figure below A and Figure EV3 I). In a corresponding analysis, comparing single knockdown of *Prdm16* with the double knockdown, there is no significant similarity (at $\alpha=0.05$ or $\alpha=0.1$) (Figure below B), neither was there a significant difference between the groups (Fig. 5 H).

Figure legend-Equivalence test for single and double Knockdown of *Ctcflos* and *Prdm16*. Paired TOST-Equivalence test comparing *Ucp1* transcript levels of (A) *Ctcflos* single with *Ctcflos+Prdm16* double KD and (B) *Prdm16* single with *Ctcflos+Prdm16* double KD. Significance level alpha is set to 0.1. Equivalence bounds were set based on the multiplicative gene interaction model.

In summary, investigating the epistasis between *Ctcflos* and *Prdm16* in the regulation of *Ucp1* gene expression we made the following observations:

I) The *Ctcflos* knockdown effect on *Ucp1* expression can be rescued by *Prdm16* overexpression, indicating that the *Ctcflos* pathway in brite adipogenesis involves *Prdm16* as downstream mediator.

II) *Ucp1* expression is similar (at $\alpha=0.1$) between *Ctcflos* single and *Ctcflos+Prdm16* double knockdown cells, supporting the absence of a considerably large additive effect of the combined knockdown, further supporting that the *Ctcflos* pathway in brite adipogenesis involves *Prdm16* as downstream mediator.

III) There is neither a significant similarity nor a significant difference between *siPrdm16+Ctcflos* ASO1 and the individual KD of *Prdm16*. The lack of similarity between these two groups actually provides a notable hint towards a broader, more complex regulatory activity of *Ctcflos* in brite adipogenesis that includes but also goes beyond the regulation of *Prdm16*. The by trend, though not significantly, stronger impact of the double KD points towards additional mechanistic activities of *Ctcflos*, which is in accordance with its parallel implication in alternative splicing modulation.

Together, these observations strengthen the epistasis between *Ctcflos* and *Prdm16*, pointing toward a complex regulatory function of *Ctcflos* that partially depends on transcriptional modulation of *Prdm16* and entails further mechanistic routes.

5. Throughout the paper there are various changes shown, and for some of them its not clear if they are significant or not, although they seem to be treated as such. For example Fig. 4L, 4S We thank the reviewer for this note and addressed it by clarifying the significance of all displayed changes. In the first version of this manuscript all significant changes were indicated by stars, the lack of stars indicated a non-significant change. We agree that this might raise the question whether there was statistical testing or not. To clarify this, we now indicated all non-significant changes by the abbreviation “n.s.”. Accordingly we included indications of significance in graphs Fig. 3 E, Fig. EV 1 M (previously Fig. 4 L), Fig. 4 O and P (previously Fig. 4 R and S), Fig. 6 B, L, M, P, U (previously Fig. 6 B, J, K, Fig. S 7 C and F), Fig EV1 O (previously Fig. S 3), Fig. EV 7 D, F, G (previously Fig. S 7 K, M, N).

The changes in previous Fig. 4 L (Fig. 1 M in the revised manuscript) are not significant. To make this more clear we rephrased lines 266-271 in the following way: “Knockdown of *Ctcflos* tended to shift cell morphology towards reduced brite characteristic multilocularity, with slightly lower lipid droplet numbers (Fig. EV 1 J). **The observed trend for increased lipid droplet sizes was not significant (Fig EV1K-M).** *Ctcflos* deficient cells maintained the ability to differentiate into lipid loaded adipocytes, as indicated by the lack of significant reduction in Oil red O staining (Fig EV1N and O).” (lines 272-275 in the revised manuscript)

Due to the lack of a significant effect on lipid droplet size and according to the suggestion of Referee #2 we moved the figure panel to the supplement (Fig. EV 1 K-M in the revised manuscript; previously Fig. 4 J, K, L).

The changes in the lipolysis rates (previously Fig. 4 S; Fig. 4 P in the revised manuscript) in *Ctcflos* control and KD cells, are also not significant. This is stated in the results part of the manuscript: line 285 in the revised manuscript: “Since lipolysis was **not reduced** in isoproterenol-stimulated *Ctcflos* deficient cells...”.

Minor comments:

1. Line 173: The finding that lncRNAs are differentially expressed does not support the "initial hypothesis about their regulatory involvement". Genes can be differentially expressed yet be completely uninvolved in regulation / non-functional. This statement should be toned down. Same for line 193: "substantiated the involvement of positively regulated lncRNAs in the browning process".

We agree with the reviewer that differential expression of a gene during differentiation does not necessarily mean that it is functionally involved in the regulation of this process. We therefore rephrased the sentence “... supporting our initial hypothesis about their regulatory involvement in the process of brite cell differentiation” into: “... **demonstrating their transcriptional regulation during brite cell differentiation. Among them there might be lncRNAs that are functionally involved in the regulation of this process.**” (lines 160-162 in the revised manuscript)

The sentence “This further substantiated the involvement of positively regulated lncRNAs in the browning process” was changed to “, **proposing a similarly important role of this group of lncRNAs in the browning process.**” (lines 180-181 in the revised manuscript)

2. Is there any evidence that *Ctcflos* is conserved in human? This should be mentioned/discussed.

In order to examine whether murine lncRNA *Ctcflos* is conserved in the human genome, we searched for lncRNA annotations located between *Ctcf1* and *Pck1*, the up- and downstream neighbor genes of *Ctcflos* in the murine locus. FANTOM (Functional Annotations of the

Mammalian Genome) predicted one human lncRNA upstream of *Pck1* (Figure below). Alignment of the human lncRNA sequence against all murine *Ctcflos* transcripts, however, did not show considerable matches (Analysis performed by Dr. Juan Carlos Higareda Almaraz). Similarly, alignment of the murine *Ctcflos* transcript sequences against the human genome and transcriptome by ensemble BLAT and BLASTN, respectively, did not reveal alignments of considerable lengths. Based on these results we conclude that *Ctcflos* is not conserved in human. We included this information in the manuscript (lines 232-233 in the revised manuscript): “The lack of human orthologs of the four murine *Ctcflos* transcripts indicates that it is not conserved in human.”

Figure legend - Lack of a human *Ctcflos* ortholog. (A) Human gene locus including *Cctf1*, *Pck1* and a human lncRNA encoded between these loci. (B) Alignments of the in A highlighted human lncRNA with the sequences of the murine *Ctcflos* transcripts.

Referee #2:

The authors propose the lncRNA *Ctcflos* as a regulator of adipose browning, and perform an impressive number of experiments and analysis to characterize its function within this context. Their results suggest that the main role of *Ctcflos* in browning is through regulation of expression and alternative splicing of *Prdm16*, a well known regulator of brown adipocyte development.

This work is a part of the important but gigantic task of characterizing the function of non-coding RNA in the mammalian genome. Although several efforts have been made to establish function for non-coding RNA at a genome wide level, it is becoming increasingly clear that these efforts need to be complemented by deep studies of individual loci in specific cellular contexts. The authors present here one example of this laborious but much needed work.

The manuscript also illustrates the difficulties involved in determining non-coding RNA function. These transcripts are usually lowly expressed, unstable, and may exist in many unannotated forms. Designing appropriate knock-down or over-expression experiments is not an easy task and there are many pitfalls.

The manuscript is clear and easy to follow. Most conclusions are supported by data but there is a design flaw in the ASO experiments: relying on only one ASO per isoform when knocking down *Ctcflos*. I detail my concerns below.

Major:

1. A design flaw in the study is that only one ASO was used for knock-down of each isoform of *Ctcflos*. When using oligo based knock-down (ASOs, shRNA, siRNA etc) it is widely recommended to use multiple independent oligos targeting the molecule of interest, in order to minimize the risk that the observed effects of the knock-down wholly or partly result from off-target effects rather than the actual knock-down. In the current study, a better design would have been to use at least one more ASO per isoform in parallel, in order to exclude this possibility. The problem is somewhat mitigated in some experiments where two different isoforms ("tr1" and "tr3") are knocked down by different ASOs ("ASO1" and "ASO2" respectively), and it is promising that the two different ASOs seem to yield (mostly) similar results. However, for many experiments the authors abandon this approach and rely solely on results using ASO1. That leaves the possibility that some of the following observations are due to off-target effects. Best would be to redo these experiments with a different ASO, but at the very least, text should be added to the manuscript that clearly discusses the possibility that some of the ASO knockdown results may be due to off-target effects.

We thank the reviewer for this valuable comment and clearly agree with the recommendation to use multiple different antisense oligos (ASOs) to knockdown a target molecule in order to exclude or reduce the risk of off-target effects. We therefore repeated the knockdown of *Ctcflos* tr1 with four further ASOs, two targeting *Ctcflos* tr1 (ASO 3 and 4) and two targeting *Ctcflos* tr3-4 (ASO 5 and 6). Comparable to ASO 1 and 2 *Ctcflos* knockdown by the four new ASOs comparably resulted in impaired brite adipogenesis as shown by reduced *Ucp1* gene expression (Figure below) (revised manuscript Fig. 4 C-E).

Figure legend-Knockdown of *Ctcfls* transcript 1, 3 and 4 by further LNA Gapmer ASOs. (A, B) Efficiency of *Ctcfls* KD (A) by new ASOs 3 and 4 alongside previously shown ASO1 targeting *Ctcfls* transcript 1 and (B) by new ASOs 5 and 6 alongside previously shown ASO2 targeting *Ctcfls* transcript 3 and 4. (C) Impact of *Ctcfls* KD on *Ucp1* gene transcription including new and previously characterized ASOs.

For the key experiments of the study, including *Ucp1* gene expression, brite adipocyte marker gene transcription, global transcriptome analysis, lipolysis rates and brite cell respiration, we used two antisense oligos (ASO 1 and ASO 2). To complete this and to reduce the risk of off target effects of using only one antisense oligo, we further repeated the analysis of *Ctcfls*-dependent alternative splicing of *Prdm16* with ASO 2 mediated *Ctcfls* KD, providing similar results as using ASO 1 (Figure below) (revised manuscript Fig. 6 O-X).

Figure legend-Impact of *Ctcfls* KD on *Prdm16* alternative splicing evaluated qPCR using ASO1 and ASO2. (A) Splice graph of *Prdm16* and resulting *Prdm16* long and short isoforms. Primers used to amplify specifically *Prdm16* short and long isoforms in qPCR are displayed in blue. (B-K) Effect of *Ctcfls* KD on *Prdm16* isoform expression evaluated by qPCR. (B-F) Relative *Prdm16* total, short and long transcript levels after 24 hours in response to *Ctcfls* KD by ASO1 and ASO2 compared to nontargeting controls. (E, F) Comparison of fold changes after 24 hours between control and *Ctcfls* KD cells ((E) KD by ASO1, (F) KD by ASO 2) for *Prdm16* short and long isoform, respectively. (G-K) Relative *Prdm16* total, short and long transcript levels after 72 hours in response to *Ctcfls* KD by ASO1 and ASO2 compared to nontargeting controls. (J, K) Comparison of fold changes after 72 hours between control and *Ctcfls* KD cells ((J) KD by ASO1, (K) KD by ASO 2) for *Prdm16* short and long isoform, respectively. Mean values \pm SD, n=3-4, unpaired t tests, n.s. p>0.05, *p<0.05, **p<0.01 ***p<0.001.

2. The authors conclude from their experiments that *Ctcfls* has a dual function: modulating the transcription levels as well as isoform ratios of *Prdm16*. But whereas possible mechanisms through which *Ctcfls* may influence the splicing machinery are discussed at length in the Discussion section, no similar suggestions for how *Ctcfls* may regulate transcript levels are

introduced in the manuscript. In fact, the experiments are entirely consistent with a less complicated explanation: that altered transcription levels follows from altered splicing. Unless evidence is presented that there are two distinct mechanisms (or at least extensive discussion detailing this reasoning), it seems unwarranted to put so much emphasis on this dual function hypothesis.

As the reviewer pointed out, in the first version of the manuscript we described *Ctcflos* to exert a dual functionality in brite adipogenesis by modulating transcription and alternative splicing. Characterizing *Ctcflos* KD adipocytes revealed altered transcript levels as well as shifted splicing patterns of several genes of the thermogenic program, mitochondrial function and lipid metabolism. For both, the regulation of alternative splicing and the regulation of transcription, several putative mechanisms have been described how lncRNAs can be involved: lncRNAs can interact with the pre-mRNA itself, RNA binding proteins or splicing factors to modulate the splicing process. By interaction with chromatin modifiers, transcription factors or cofactors to guide them to or hijack them from their sites of transcription lncRNAs can regulate transcription of one or several target genes.

Since altered transcription and splicing occur in parallel as direct consequence of the knockdown and we do not have information on the detailed mechanistic activity and direct target of *Ctcflos* that mediates altered transcription and splicing, it is indeed difficult to disentangle the causal chain of cellular changes that occur in response to *Ctcflos* KD. It is of course possible that altered splicing, especially of key browning factors such as *Prdm16*, can entail extensive transcriptional changes as observed in the present study. On the other hand the *Ctcflos* KD dependent transcriptional modifications could likewise be causal for the splicing changes (e.g. by transcriptional changes of the splicing machinery and of splicing factors). In a third scenario altered transcription and alternative splicing might occur in parallel, not necessarily mediated by two independent mechanistic activities of *Ctcflos* but maybe rather as a co-regulated process. Our data provide some hints that favor this third scenario: First, the *Ctcflos* KD-induced changes in transcription and alternative splicing occur synchronously after the KD. In our two-time point RNA sequencing strategy that was intended to help distinguish cause and consequence, transcription and splicing changes both occurred 24 hours after *Ctcflos* KD. If one would occur as consequence of the other they should appear in chronological succession. Further arguing for a co-regulation of the two processes is the fact that transcriptional and splicing changes do not strictly target entirely distinct sets of genes but several genes are both, changed in transcription level and alternative splicing pattern as showcased by *Prdm16*. In fact, transcription and splicing are tightly associated processes that are spatially and temporally coordinately regulated in the nucleus. Prior studies described that transcription factors and RNA polymerase II can directly or indirectly (through cofactors (such as PGC1 α)) interact with splicing factors to concomitantly modulate the transcription rate and splicing profile of a target gene. (Auboeuf *et al*, 2004; Cramer *et al*, 1999; Das *et al*, 2007; Monsalve *et al*, 2000). Interestingly, nuclear speckles that function as storage, assembly and modification compartments of transcription and splicing associated factors play a major role in the coordination of both processes and can be regarded as kind of a common denominator of transcription and splicing regulation. Phosphorylation and dephosphorylation cycles of splicing factors, transcription factors and cofactors by kinases and phosphatases within the speckles coordinate their activity, assembly to higher order complexes and translocation to the site of transcription/RNA processing. These processes in turn can be regulated by lncRNAs (as previously shown for *Malat1* (in human cells)) (Tripathi *et al*, 2010). The increased number of nuclear speckles in *Ctcflos* deficiency that might argue for an impaired translocation of activated transcription and splicing associated factors out of the speckles to their site of action suggest the involvement of *Ctcflos* in this regulatory process. An interaction of *Ctcflos* with transcription factors, coregulators, splicing factors or kinases and proteases within nuclear

speckles to modulate their phosphorylation (and thus activity) states, their translocation or assembly to complexes could be envisioned but still awaits experimental exploration. To further clarify this hypothesis subnuclear localization of *Ctcflos* (e.g. by RNA FISH) and *Ctcflos* interaction partners shall be identified in future analyses.

Conclusively, due to the lack of any clear evidence of cause and consequence and the hint towards *Ctcflos*-dependent nuclear speckle function, we would prefer to describe the transcription and splicing changes as parallel occurring effects of *Ctcflos* KD and would like to rise the hypothesis that *Ctcflos* coordinates these processes as component of the nuclear speckle machinery where transcription and alternative splicing factors are commonly regulated.

As pointed out by the reviewer this is clearly not sufficiently discussed in the manuscript. The discussion was adjusted to make this point more clear (lines 601-623 in the revised manuscript). Also, the term “bi-functionality” was revised as it might be misleading to the impression that *Ctcflos* works through two independent mechanisms for which we do not present any evidence and which we did not intend to indicate (lines 40, 478, 492, 653). We rather aimed to describe that *Ctcflos* exerts an impact on two levels of brite gene expression, on transcription and alternative splicing.

Minor:

* In Fig 1B, I don't understand the meaning of the legend: what is "Contrib" and "Condition"? The term “Condition” in the figure legend of principal component analysis refers to the two different groups of samples, undifferentiated (“undiff”) and differentiated (“diff”) adipocyte samples of the five mouse strains, respectively. To clarify this, we renamed it “**stage of adipocyte differentiation**”. The term “Contrib” that also discriminates between “undiff” and “diff” is redundant and does not provide further information. We apologize for this mistake and changed the figure legend (Fig. 1 B).

* Fig 4J,K,L, the results regarding lipid droplet size: are any differences between NC and ASO1 KD significant? If not, these figures and corresponding text could be dropped or moved to supplement.

The lipid droplets of *Ctcflos* KD adipocytes only show a trend towards reduced sizes but are not significantly changed, we therefore moved, as suggested, the figure panel to the supplement (previous Fig. 4 J, K, L are now Fig. EV 1 K, L, M).

Additionally, to make more clear that there is no significant effect, we rephrased lines 271-275 in the following way: “Knockdown of *Ctcflos* tended to shift cell morphology towards reduced brite characteristic multilocularity, with slightly lower lipid droplet numbers (Fig. EV 1 J). **The observed trend for increased lipid droplet sizes was not significant (Fig. EV 1 K-M).** Despite this, *Ctcflos* deficient cells maintained the ability to differentiate into lipid loaded adipocytes, as indicated by the lack of significant reduction in Oil red O staining (Fig. EV 1 N, O).”

* Fig 5E, what was used as the background gene set in GO analysis? All genes, or all genes expressed in adipose tissue, or something else?

The background gene set for GO-term analysis by InCroMap software included all murine genes.

* Fig 5F, it is not easy to understand how to understand how to interpret the "Path Visio" figure and what it actually tries to visualize.

The Path Visio figure visualizes the *Ctcflos* knockdown induced transcriptional changes of the components of the core thermogenic gene expression program. The figure summarizes

important transcription regulatory events that drive precursor cell commitment into thermogenic preadipocytes as well as early and late differentiation into mature thermogenic adipocytes. In the commitment phase, PRDM16 in collaboration with Euchromatic histone-lysine N-methyltransferase 1 (EHMT1) or C-Terminal Binding Protein 1 (CTBP1) and further histone modifying enzymes represses the transcription of muscle and white fat selective genes. In a complex with PPAR γ , TATA-binding protein associated factor 7L (TAF7L) and transcription factor IID (TFIID), PRDM16 is further involved in the transcriptional activation of thermogenic adipocyte-selective genes to drive cell fate into the thermogenic direction. During early differentiation transcription factors, including CEBP β , EBF2, PPAR γ together with its coactivator PGC1 α as well as ZFP516 bind to enhancer and promoter elements of thermogenic genes. *Prdm16* associates with these factors and recruits the Mediator complex to drive transcription of the targeted genes. Similarly, *Ucp1* gene transcription is mediated in the late differentiation phase by the interplay of several transcription factors and coactivators, giving rise to functional thermogenic adipocytes. **Fig. 5 F** summarizes these events and shall introduce those readers not familiar to thermogenic adipogenesis with the most important players in this procedure. Additionally, it provides information about the influence of *Ctcflos* on the transcription of these factors. The color code visualizes the log₂ fold change of genes 24 hours (left part of box) and 72 hours (right part of box) after *Ctcflos* KD. Several transcription factors and cofactors are transcriptionally downregulated at one or both time points, indicating that *Ctcflos* KD affects the core thermogenic transcription regulation program. Importantly, the discrimination between early (24 hours) and later (72 hours) KD induced changes provides significant information on the sequential order of *Ctcflos* KD events, revealing more proximal (*Prdm16*, *Ebfl*, *Ppar γ* , *Nfia*) and more distant (*Pgc1 α* , *Ppara*, *Thr β* , *Cebp β*) mediators of the *Ctcflos* KD impact on *Ucp1* gene transcription.

* Fig 6A, why did these terms not show up among terms discovered in Fig 5E? Or Fig 5E only contains a selected set of GO terms? Please clarify.

Figure 5 E presents the GO term analysis of genes that were **DOWN**-regulated 72 hours after *Ctcflos* knockdown, while **Figure 6A** shows the GO term analysis of genes that were **UP**-regulated 24 hours after *Ctcflos* knockdown as well as the corresponding GO-terms in the analysis of UP-regulated genes 72 hours after *Ctcflos* knockdown. Different sets of regulated genes (up vs. down regulated) were thus subjected to the analyzes in Figures 5 and 6, explaining why there is no overlap in the overrepresented GO terms in both Figures. We stated that more clearly in the revised Manuscript line 393.

Figure 5 E and Figure 6 A (genes regulated 24 hours after KD) list the top GO terms, ranked by the strength of overrepresentation. The top ranking GO terms of the 24 hours regulated genes in Figure 6 A were extracted from the 72 hour analysis for comparison. The actual ranking position of the GO terms within the 72 hour analysis is stated by the number right next to the bar of the (-log₁₀) p-value.

* As far as I can see, Fig. 6 I is not referred to in the text.

We thank the reviewer for the note that we missed to refer to **previous Figure 6 I (revised manuscript Fig 6K)** in the text. We added the reference to the figure in line 424.

Referee #3:

Bast-Habersbrunner et al. introduce the lncRNA *Ctcflos* as novel regulator of brite adipogenesis in vitro. Using big data analysis of lncRNA-expression profiles of adipose tissues from several inbred mouse strains with varying degrees of adipose browning *Ctcflos* emerges as a candidate of interest. *Ctcflos* is shown to affect adipocyte differentiation on the levels of transcription and alternative splicing, exemplified by *Ucp1* and *Prdm16* mRNAs. Furthermore, the authors demonstrate altered adipocyte function upon knockdown of *Ctcflos*.

The research is novel, technically sound and the manuscript is very well prepared. To my opinion, this manuscript has no major weaknesses.

Minor Points:

1. This work originates from the study of inguinal white adipose tissue, which is an adipose depot that readily undergoes adipose browning. Hence, it is logical that the authors chose to study the role of *Ctcflos* in what they refer to as brite adipocytes. However, in the end, the data also allow drawing the conclusion that also in classical brown adipocytes, e.g. derived from the interscapular depot, *Ctcflos* is expressed, regulated and has functional impact. To my opinion, this could be highlighted better.

We greatly thank the reviewer for this comment and agree that the effect of *Ctcflos* in brown adipocyte differentiation and function can be strengthened in the manuscript. At different points in results and discussion we emphasized that *Ctcflos*, beside its function in brite adipocytes, also regulates classical brown adipogenesis:

- line 306-307: “Together, our data validate *Ctcflos* as an essential component in the specific control of thermogenic programming in **brite and brown** adipocytes.”
- line 496-497: “**Consistent with this pivotal role in the thermogenic gene program, *Ctcflos* likewise presents a significant component in classical brown adipogenesis.**”
- line 523-525: “Eliminating *Ctcflos* during this early phase in the thermogenic differentiation program induces severe impairments of brite **and brown** adipocyte development as evidenced on transcriptional and functional levels.”
- line 646-648: “Conclusively, our comparative transcriptome analyses of diverging iWAT browning propensities among inbred mouse strains uncovered adipose tissue enriched lncRNA *Ctcflos* that orchestrates early gene programming towards brite **and brown** adipogenesis.”
- line 653-654: “As versatile regulator, *Ctcflos* accomplishes an essential role in brite **and brown** fat cell differentiation, mitochondrial biogenesis and thermogenesis.”

2. The references to Figure S4P&Q in lines 316-323 is missing.

We apologize for the mistake. We included the references to **previous Fig. S4 P (revised manuscript Fig. EV2 T) in line 332 and previous Figure S4 Q (revised manuscript Fig. EV2 U) in line 333**

3. Figure 6B shows a decrease of *Ucp1* mRNA per *Fabp4* upon isoginkgetin: it would be nice to also show the levels of both genes in comparison to a housekeeping gene.

Showing the levels of *Ucp1* and *Fabp4* relative to the housekeeper *Gtf2b*, respectively would indeed be further informative (**Figure below B and C**). We therefore included the following data in the manuscript as **Figure 6 C and D in the revised manuscript**.

Figure legend – Impact of splicing inhibition by different concentrations of isoginkgetin on *Ucp1* mRNA levels relative to *Fabp4* and *Ucp1* and *Fabp4* mRNA levels relative to *Gtf2b*. (A) *Ucp1* transcript levels relative to *Fabp4*, (B) *Ucp1* transcript levels relative to *Gtf2b* and (C) *Fabp4* transcript levels relative to *Gtf2b*. Mean values \pm SD, n=3, one-way ANOVA (Šídák-test), n.s. $p > 0.05$, * $p < 0.05$, ** $p < 0.01$, *** $p < 0.001$.

4. In line 807 it appears that blocking was done with TritonX-100 and not "TiritonX-100" We corrected that spelling mistake in line 804 in the revised manuscript. TiritonX-100 was replaced by TritonX-100.

5. The legend in Figure 1B (not the figure legend) does not explain well what is actually shown in that PCA plot.

We apologize for the lack of clarity concerning the PCA legend. The term “Condition” refers to the two different groups of samples, undifferentiated (“undiff”) and differentiated (“diff”) adipocyte samples of the five mouse strains, respectively. To clarify this, we renamed it “**stage of adipocyte differentiation**”. The term “Contrib” that also discriminates between “undiff” and “diff” is redundant and does not provide further information. We apologize for this mistake and changed the figure legend.

6. In Figure 4F please specify the loading control of the Western Blot shown in the figure legend.

We specified β -Actin (β -Actin) as loading control within previous Figure 4 F (revised manuscript Fig. 4 G) and also included this information (“Actin- β as loading control”) in the figure legend for Figure 4 G and also for Figure 4 Q in the revised manuscript.

7. The panels in Figure 6C should be enlarged for better visibility.

We enlarged the panels in previous Figure 6 C (revised manuscript Fig. 6 E).

References

- Auboef D, Dowhan DH, Kang YK, Larkin K, Lee JW, Berget SM, O'Malley BW (2004) Differential recruitment of nuclear receptor coactivators may determine alternative RNA splice site choice in target genes. *Proceedings of the National Academy of Sciences of the United States of America* 101: 2270-2274
- Costanzo M, Kuzmin E, van Leeuwen J, Mair B, Moffat J, Boone C, Andrews B (2019) Global Genetic Networks and the Genotype-to-Phenotype Relationship. *Cell* 177: 85-100
- Cramer P, Cáceres JF, Cazalla D, Kadener S, Muro AF, Baralle FE, Kornbliht AR (1999) Coupling of transcription with alternative splicing: RNA pol II promoters modulate SF2/ASF and 9G8 effects on an exonic splicing enhancer. *Molecular cell* 4: 251-258
- Das R, Yu J, Zhang Z, Gygi MP, Krainer AR, Gygi SP, Reed R (2007) SR proteins function in coupling RNAP II transcription to pre-mRNA splicing. *Molecular cell* 26: 867-881

Goldstein LD, Cao Y, Pau G, Lawrence M, Wu TD, Seshagiri S, Gentleman R (2016) Prediction and Quantification of Splice Events from RNA-Seq Data. *PLoS one* 11: e0156132

Lundh M, Plucińska K, Isidor MS, Petersen PSS, Emanuelli B (2017) Bidirectional manipulation of gene expression in adipocytes using CRISPRa and siRNA. *Molecular metabolism* 6: 1313-1320

Monsalve M, Wu Z, Adelmant G, Puigserver P, Fan M, Spiegelman BM (2000) Direct coupling of transcription and mRNA processing through the thermogenic coactivator PGC-1. *Molecular cell* 6: 307-316

Tripathi V, Ellis JD, Shen Z, Song DY, Pan Q, Watt AT, Freier SM, Bennett CF, Sharma A, Bubulya PA *et al* (2010) The nuclear-retained noncoding RNA MALAT1 regulates alternative splicing by modulating SR splicing factor phosphorylation. *Molecular cell* 39: 925-938

Dear Dr. Klingenspor,

Thank you for submitting your revised manuscript. It has now been seen by two of the original referees.

As you can see, the referees find that the study is significantly improved during revision and recommend publication. Before I can accept the manuscript, I need you to address the points below:

- We note that Grant # LI 3716/1-1 is missing from the manuscript submission system (eJP). Moreover, in the manuscript text, the funding information should only be listed in the Acknowledgements section.
- We noticed that Figures 6F,N and Fig EV1P are currently not called out in the text.
- Please convert Table EV1 into a Dataset file. Also, its legend needs to be removed from the word article file and added into the dataset file.
- Please merge the source data into a single file.
- Please rename the 'Methods' section as 'Materials and Methods'.

Thank you again for giving us to consider your manuscript for EMBO Reports, I look forward to your minor revision.

Kind regards,

Deniz Senyilmaz Tiebe

--

Deniz Senyilmaz Tiebe, PhD
Editor
EMBO Reports

Referee #1:

The authors have addressed the comments from the previous round of review in a satisfactory manner. I now recommend publication in EMBO Reports.

Referee #3:

The authors have clarified all my points and - to my opinion - also to the other reviewers' points. Great job!

Prof. Dr. Martin Klingenspor | Technische Universität München |
LS Mol. Ernährungsmedizin | Gregor-Mendel-Str. 2 | 85354 Freising

Freising, 21.04.2021

Dear Dr. Deniz Senyilmaz Tiebe,

we greatly thank you for informing us about the reviewers positive feedback on our revision. As specified below, we tried to fulfill your additional requests and changed the manuscript, the files and the submission form accordingly. We were not completely sure about your request to convert Table EV1 into a dataset. Please let us know if there is still need for change.

- We note that Grant # LI 3716/1-1 is missing from the manuscript submission system (eJP). Moreover, in the manuscript text, the funding information should only be listed in the Acknowledgements section.
 - ➔ The Grant LI 3716/1-1 was now correctly included in the submission system
 - ➔ The extra Funding section was deleted in the manuscript, funding information are now only mentioned in the Acknowledgement section.
- We noticed that Figures 6F,N and Fig EV1P are currently not called out in the text.
 - ➔ We added the reference to Fig 6F in line 412, the reference to Fig 6N can be found in line 443, the reference to Fig EV1P can be found in line 297.
- Please convert Table EV1 into a Dataset file. Also, its legend needs to be removed from the word article file and added into the dataset file.
 - ➔ The excel file of Table EV1 has been renamed into Dataset EV1 and is added as a dataset file into the submission system. In the manuscript text "Tab EV1" was replaced by "Dataset EV1". The data summarized in the table originate from the following dataset (*Array Express E-MTAB-8344, 2019*) that is included in the reference list. We added the reference to this Dataset in line 185-186 where Tab. EV1 is named to clarify this. We hope these changes fulfill your request.
 - ➔ We removed the dataset legend from the Manuscript_Text word file. The legend is included in the Dataset EV1 excel file.
- Please merge the source data into a single file.
 - ➔ All source data have been merged into one jpg file
 - ➔ Alternatively, all source data files have been merged into a single .zip file Both versions are now included in the submission form
- Please rename the 'Methods' section as 'Materials and Methods'.
 - ➔ The Methods section has been renamed into 'Material and Methods'

We are looking forward to hearing from you.

Sincerely,

Prof. Dr. Martin Klingenspor

Technical University of Munich
TUM School of Life Sciences
Weihenstephan
Chair of Molecular Nutritional Medicine

EKFZ – Else Kröner-Fresenius Zentrum für
Ernährungsmedizin

ZIEL – Institute for Food & Health

Dear Prof. Klingenspor,

Thank you for submitting your revised manuscript. I have now looked at everything and all is fine. Therefore, I am very pleased to accept your manuscript for publication in EMBO Reports.

Congratulations on a nice work!

Kind regards,

Deniz Senyilmaz Tiebe

--

Deniz Senyilmaz Tiebe, PhD
Editor
EMBO Reports

--

Please note that under the DEAL agreement of German scientific institutions with our publisher Wiley, you could be eligible for free publication of your article in the open access format. Please contact either the administration at your institution or our publishers at Wiley (emboreports@wiley.com) for further questions.

At the end of this email I include important information about how to proceed. Please ensure that you take the time to read the information and complete and return the necessary forms to allow us to publish your manuscript as quickly as possible.

As part of the EMBO publication's Transparent Editorial Process, EMBO reports publishes online a Review Process File to accompany accepted manuscripts. As you are aware, this File will be published in conjunction with your paper and will include the referee reports, your point-by-point response and all pertinent correspondence relating to the manuscript.

If you do NOT want this File to be published, please inform the editorial office within 2 days, if you have not done so already, otherwise the File will be published by default [contact: emboreports@embo.org]. If you do opt out, the Review Process File link will point to the following statement: "No Review Process File is available with this article, as the authors have chosen not to make the review process public in this case."

Should you be planning a Press Release on your article, please get in contact with emboreports@wiley.com as early as possible, in order to coordinate publication and release dates.

Thank you again for your contribution to EMBO reports and congratulations on a successful publication. Please consider us again in the future for your most exciting work.

THINGS TO DO NOW:

You will receive proofs by e-mail approximately 2-3 weeks after all relevant files have been sent to

our Production Office; you should return your corrections within 2 days of receiving the proofs.

Please inform us if there is likely to be any difficulty in reaching you at the above address at that time. Failure to meet our deadlines may result in a delay of publication, or publication without your corrections.

All further communications concerning your paper should quote reference number EMBOR-2020-51289V3 and be addressed to emboreports@wiley.com.

Should you be planning a Press Release on your article, please get in contact with emboreports@wiley.com as early as possible, in order to coordinate publication and release dates.

Corresponding Author Name: Prof. Dr. Martin Klingenspor

Manuscript Number: EMBOR-2020-51289V1